# Microtubule-mediated GLUT4 trafficking is disrupted in insulin-resistant skeletal muscle

Jonas R Knudsen[1,2]*, Kaspar W Persson[1], Carlos Henriquez-Olguin[1,3], Zhencheng Li[1,4], Nicolas Di Leo[1], Sofie A Hesselager[1], Steffen H Raun[5], Janne R Hingst[6], Raphaël Trouillon[7], Martin Wohlwend[8], Jørgen FP Wojtaszewski[1], Martin AM Gijs[9], Thomas Elbenhardt Jensen[1]*

[1]August Krogh Section for Molecular Physiology, Department of Nutrition, Exercise and Sports, Faculty of Science, University of Copenhagen, Copenhagen, Denmark; [2]Heart and Skeletal Muscle Biology, Global Drug Discovery, Novo Nordisk, Soeborg, Denmark; [3]Exercise Science Laboratory, Faculty of Medicine, Universidad Finis Terrae, Santiago, Chile; [4]College of Physical Education, Chongqing University, Chongqing, China; [5]Department of Biomedical Sciences, Faculty of Health and Medical Sciences, University of Copenhagen, Copenhagen, Denmark; [6]Clinical Drug Development, Novo Nordisk, Soeborg, Denmark; [7]Department of Electrical Engineering, Polytechnique Montréal, Montréal, Canada; [8]Laboratory of Integrative Systems Physiology, Institute of Bioengineering, École Polytechnique Fédérale de Lausanne, Lausanne, Switzerland; [9]Microsystems Laboratory 2, Institute of Electrical and Micro Engineering, École Polytechnique Fédérale de Lausanne, Lausanne, Switzerland

**\*For correspondence:**
jrk@nexs.ku.dk (JRK);
tejensen@nexs.ku.dk
(TElbenhardtJ)

**Abstract** Microtubules serve as tracks for long-range intracellular trafficking of glucose transporter 4 (GLUT4), but the role of this process in skeletal muscle and insulin resistance is unclear. Here, we used fixed and live-cell imaging to study microtubule-based GLUT4 trafficking in human and mouse muscle fibers and L6 rat muscle cells. We found GLUT4 localized on the microtubules in mouse and human muscle fibers. Pharmacological microtubule disruption using Nocodazole (Noco) prevented long-range GLUT4 trafficking and depleted GLUT4-enriched structures at microtubule nucleation sites in a fully reversible manner. Using a perifused muscle-on-a-chip system to enable real-time glucose uptake measurements in isolated mouse skeletal muscle fibers, we observed that Noco maximally disrupted the microtubule network after 5 min without affecting insulin-stimulated glucose uptake. In contrast, a 2-hr Noco treatment markedly decreased insulin responsiveness of glucose uptake. Insulin resistance in mouse muscle fibers induced either in vitro by C2 ceramides or in vivo by diet-induced obesity, impaired microtubule-based GLUT4 trafficking. Transient knockdown of the microtubule motor protein kinesin-1 protein KIF5B in L6 muscle cells reduced insulin-stimulated GLUT4 translocation while pharmacological kinesin-1 inhibition in incubated mouse muscles strongly impaired insulin-stimulated glucose uptake. Thus, in adult skeletal muscle fibers, the microtubule network is essential for intramyocellular GLUT4 movement, likely functioning to maintain an insulin-responsive cell surface recruitable GLUT4 pool via kinesin-1-mediated trafficking.

## Editor's evaluation

This manuscript reveals localization of Glut4 glucose transporters at microtubules in mouse and human muscle fibers and shows that disruption of microtubules or a kinesin[-1] motor alters Glut4 trafficking. Evidence is also provided supporting the idea that insulin resistance disrupts Glut4 dynamics

at microtubules. Overall, these studies provide compelling evidence that Glut4 and its regulation by insulin involves Glut4 movements that require microtubule function.

## Introduction

Skeletal muscle is quantitatively the largest site of glucose disposal, a process facilitated by insulin and contraction-responsive translocation and insertion of glucose transporter 4 (GLUT4) into the surface membrane of muscle fibers (*Jaldin-Fincati et al., 2017*; *Klip et al., 2019*). Insulin-resistant human and rodent muscle exhibit impaired insulin-stimulated GLUT4 translocation (*Zierath et al., 1996*; *King et al., 1992*; *Etgen et al., 1997*; *Garvey et al., 1998*) and muscle-specific deletion of GLUT4 is sufficient to cause systemic insulin resistance and glucose intolerance (*Zisman et al., 2000*). However, the details of GLUT4 regulation – particularly in adult skeletal muscle – and the causes of skeletal muscle insulin resistance remain unclear. In L6 myoblasts and 3T3-L1 adipocytes, insulin resistance not only decreases insulin-stimulated GLUT4 recruitment to the surface membrane, but also affects the distribution of GLUT4 between intracellular compartments (*Foley and Klip, 2014*; *Xiong et al., 2010*). This suggests that disturbed intracellular sorting of GLUT4 contributes to peripheral insulin resistance.

Motor protein-mediated trafficking on the microtubule cytoskeleton is well established to allow long-range transport of a diverse assortment of molecules and to position intracellular organelles and membrane structures in various cell types (*de Forges et al., 2012*). For GLUT4, long-range microtubule-dependent GLUT4 movement beneath the plasma membrane has been observed in adipocyte cell culture (*Lizunov et al., 2005*) and similar long-range movement was also seen in adult rodent skeletal muscle (*Lizunov et al., 2012*). A requirement for microtubule-based protein trafficking is supported by several observations in cultured cells. Microtubule disruption dispersed perinuclear GLUT4 in 3T3-L1 adipocytes (*Guilherme et al., 2000*; *Fletcher et al., 2000*) as well as L6 myoblasts (*Foley and Klip, 2014*) and impaired GLUT4 membrane insertion in some (*Foley and Klip, 2014*; *Fletcher et al., 2000*; *Chen et al., 2008*; *Emoto et al., 2001*) but not all studies (*Molero et al., 2001*; *Shigematsu et al., 2002*). Neither the requirement of microtubules for intracellular GLUT4 positioning and trafficking nor the influence of insulin resistance on microtubules and/or microtubule-based GLUT4 trafficking have been investigated in adult skeletal muscle fibers.

Therefore, we presently characterized various aspects of microtubule-based GLUT4 trafficking in predominantly adult human and mouse skeletal muscle. Our findings suggest that an intact microtubule network is required for KIF5B-mediated intracellular GLUT4 movement and maintaining insulin-responsive glucose uptake, and that impaired microtubule-based GLUT4 trafficking is a feature of skeletal muscle insulin resistance.

## Results

### GLUT4 was enriched at microtubule nucleation sites and traveled on microtubule filaments in adult mouse and human muscle

To study the involvement of microtubules in GLUT4 trafficking, we first used structured illumination microscopy to image the subsarcolemmal (up to 4 µm into the muscle fiber) microtubule network and GLUT4 in mouse and human skeletal muscle at super-resolution. Due to amenability for live fiber isolation we used flexor digitorum brevis (FDB), a muscle consisting predominantly of type IIa and IIx fibers (*Tarpey et al., 2018*), from mice and vastus lateralis, a highly mixed muscle (*Staron, 1991*; *Horwath et al., 2021*), from humans. In both mouse (*Figure 1A*) and human (*Figure 1B*) muscle, we observed GLUT4 to be localized on microtubule filaments and to be enriched at microtubule filament intersections, previously identified as microtubule nucleation sites (*Oddoux et al., 2013*). Next, to study GLUT4 movement in live muscle fibers, we overexpressed GLUT4-7myc-GFP (GLUT4-GFP) (*Bogan et al., 2001*) alone or together with mCherry-Tubulin in mouse FDB muscle fibers (*Figure 1C*). GLUT4-GFP was localized in the same pattern as endogenous GLUT4 and observed along the microtubule network, including on the more stable subpopulation (*Bulinski and Gundersen, 1991*) of detyrosinated microtubules (*Figure 1—figure supplement 1A*) implicated in trafficking of lipid droplets, mitochondria, and autophagosomes in other cells types (*Mohan et al., 2019*; *Herms et al., 2015*), as well as on mCherry-Tubulin-labeled microtubules (*Figure 1—figure supplement 1B*).

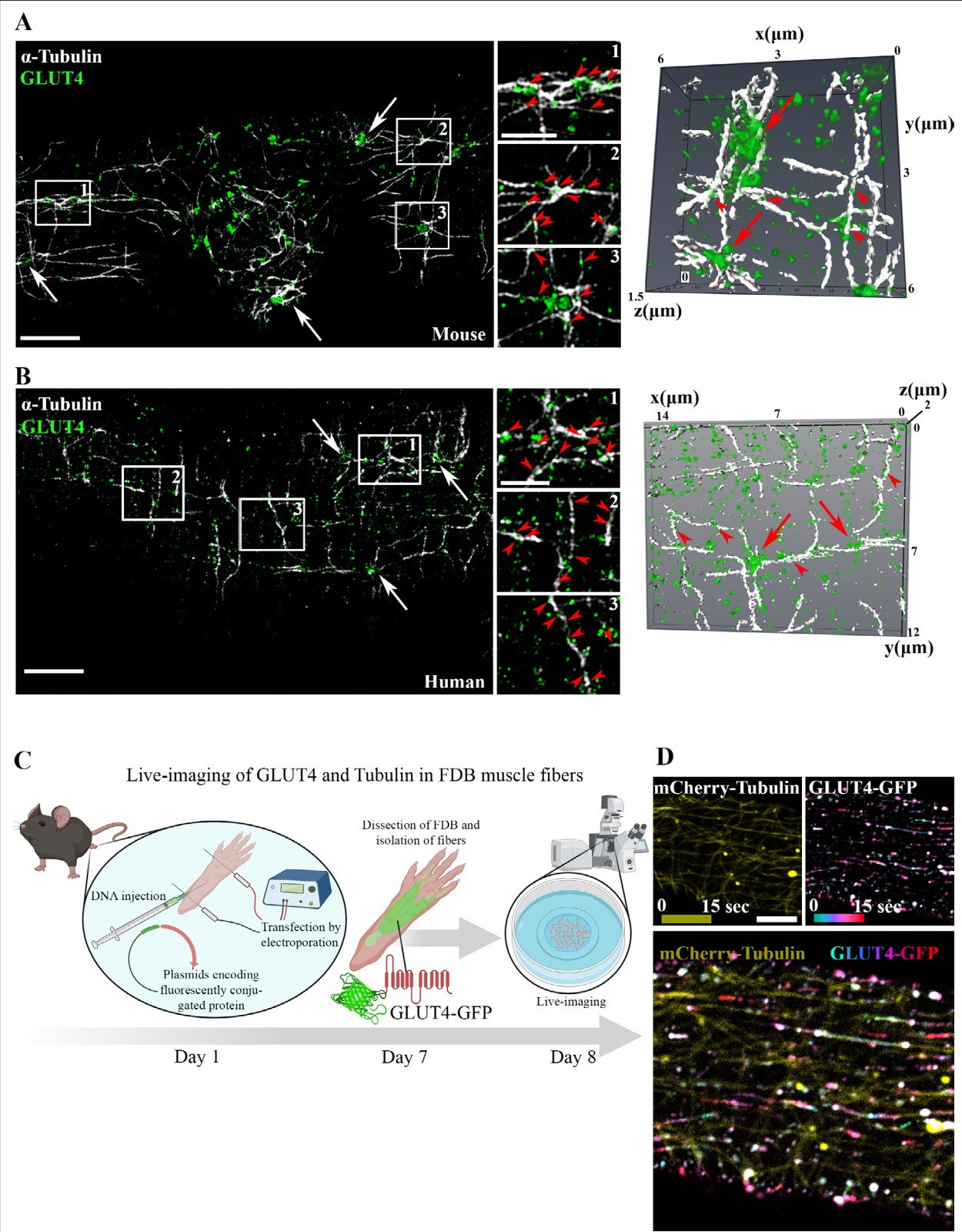

**Figure 1.** Glucose transporter 4 (GLUT4) was enriched at microtubule nucleation sites and traveled on microtubule filaments in mouse and human muscle. Structured illumination microscopy (SIM) in mouse flexor digitorum brevis (FDB) muscle (**A**) and human vastus lateralis muscle (**B**) of endogenous α-tubulin and GLUT4 (left panel) and 3D reconstruction of GLUT4 (green) and α-tubulin (white) (right panel). Arrows indicate GLUT4 at microtubule nucleation sites, arrowheads mark GLUT4 vesicles along the microtubule filaments. (**C**) Overview of workflow for live-imaging of fluorescently conjugated

*Figure 1 continued on next page*

*Figure 1 continued*

proteins in adult mouse FDB muscle fibers. (**D**) Live-imaging of FDB-expressing GLUT4-GFP and mCherry-Tubulin. Yellow projection of mCherry-Tubulin outlines the microtubule filaments (top panel left). Movement of GLUT4-GFP was visualized by color-coded projection (first image cyan, last image red, top panel right). The merged projection (bottom), demonstrated movement of GLUT4-GFP along the mCherry-Tubulin containing microtubule filaments indicated by color-coded projections on top of the microtubule filaments. The movement of GLUT4-GFP is shown in *Figure 1—video 1*. (**A, B**) Images are representative of >5 fibers from ≥3 different mice in A + D and 3 different fibers from 3 different subjects in B. Scale bar = 5 μm (**A, B, D**) and 2 μm (inserts in **B, D**).

The online version of this article includes the following video and figure supplement(s) for figure 1:

**Figure supplement 1.** Glucose transporter 4 (GLUT4) undergoes budding and fusion on the microtubules and moves bidirectionally to and from the microtubule nucleation sites.

**Figure 1—video 1.** Glucose transporter 4 (GLUT4) traveled on microtubules.

https://elifesciences.org/articles/83338/figures#fig1video1

**Figure 1—video 2.** Glucose transporter 4 (GLUT4) traveled to and from GLUT4-enriched regions at microtubule nucleation sites.

https://elifesciences.org/articles/83338/figures#fig1video2

Live-imaging revealed long-range lateral directional movement of GLUT4 along filamentous tracks (*Figure 1—figure supplement 1C*), corresponding to mCherry-Tubulin containing microtubule filaments (*Figure 1D* and *Figure 1—video 1*). The GLUT4 structures occasionally exhibited long tubular morphology (>2 μm) but were mostly minor tubular structures or spheres (size varying from ~0.4 μm$^2$ down to the unresolvable) observed to undergo budding and fusion events on the microtubule tracks (*Figure 1—figure supplement 1D, E*). Live-imaging, including fluorescence recovery after photobleaching experiments, revealed particularly dynamic and bidirectional movement at the microtubule nucleation sites (*Figure 1—figure supplement 1F–H* and *Figure 1—video 2*). Collectively, a portion of GLUT4 localized to microtubule nucleation sites and on microtubule filaments in adult mouse and human skeletal muscle. Furthermore, GLUT4 underwent continuous movement, budding and fusion along the microtubule tracks in live mouse skeletal muscle.

## GLUT4 trafficking and localization required intact microtubules

Next, we tested if microtubule-based GLUT4 trafficking was insulin responsive and dependent on an intact microtubule network. Insulin (30 nM) stimulation increased insulin signaling at the level of Akt Thr308 as expected (*Figure 2—figure supplement 1A*), and the microtubule depolymerizing compound Nocodazole (Noco) (13 μM) significantly reduced both the total and the Noco-resistant (*Khawaja et al., 1988*) detyrosinated pool of polymerized microtubules by ~90% and ~50%, respectively (*Figure 2—figure supplement 1B, C*). We did not observe any significant increase in GLUT4 movement on microtubules upon insulin stimulation but GLUT4 movement was completely prevented by microtubule depolymerization (*Figure 2A, B*, *Figure 2—figure supplement 1D*, *Figure 2—video 1*). Having established that GLUT4 trafficking was dependent on the microtubule network, we next tested if microtubule disruption affected the overall GLUT4 localization and distribution between different compartments. For quantification, we divided the GLUT4 structures into size categories corresponding to (1) large structures at the microtubule nucleation sites (>4 μm$^2$), (2) intermediate endomembrane structures (0.4–4 μm$^2$) (*Gruenberg, 2001*; *Huotari and Helenius, 2011*), and (3) the smallest resolvable endomembrane structures (<0.4 μm$^2$) including presumably insulin-responsive GLUT4 storage vesicles (GSVs) (*Figure 2—figure supplement 1E*). Microtubule disruption by Noco (13 μM) drained the GLUT4 structures at the microtubule nucleation sites and reduced the amount of the smallest structures, while causing an increase in the intermediate structures (*Figure 2C, D*). These changes were reversed within 9 hr after removal of Noco (*Figure 2C, D*). Within the smallest category, there was a shift toward fewer but larger area GLUT4 membrane structures (*Figure 2—figure supplement 1F*). The total number and area of GLUT4 structures did not differ between conditions (*Figure 2—figure supplement 1G*). In a previous study in L6 myoblasts, microtubule disruption prevented pre-internalized GLUT4 from reaching a Syntaxin6-positive perinuclear subcompartment involved in GSV biogenesis and from undergoing insulin-responsive exocytosis (*Foley and Klip, 2014*). We therefore tested in adult muscle, if microtubule disruption similarly prevented accumulation in a perinuclear Syntaxin6-positive subcompartment. However, we observed a limited and Noco-insensitive (in mice) co-localization of Syntaxin6 with either endogenous GLUT4 in human and mouse skeletal muscle, or fluorescent GLUT4-EOS (*Lizunov et al., 2013*) in mouse skeletal muscle

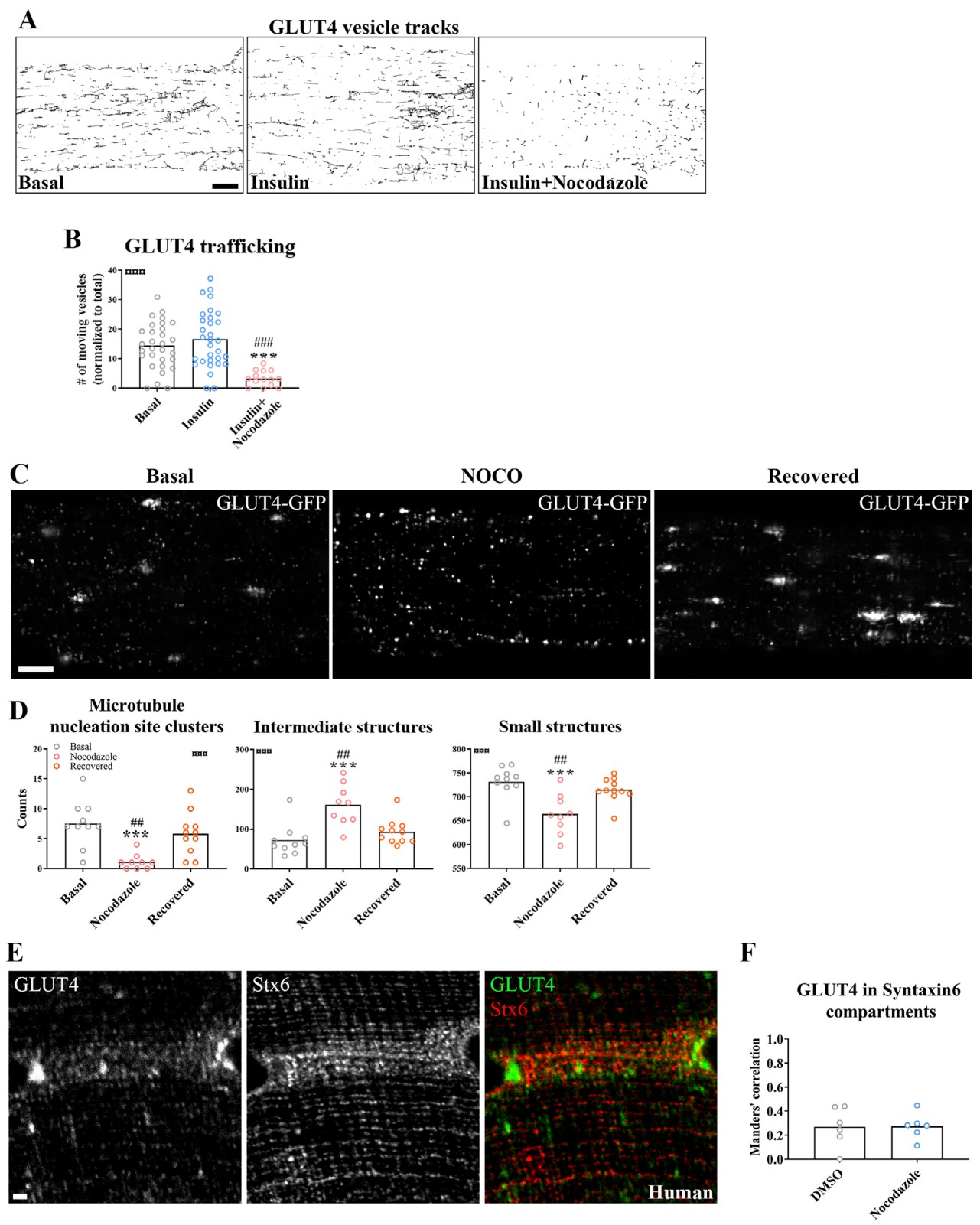

**Figure 2.** Glucose transporter 4 (GLUT4) trafficking and localization was dependent on an intact microtubule network. (**A**) Representative time-lapse traces of GLUT4-GFP vesicle tracking in muscle fibers ± insulin (INS, 30 nM) for 15–30 min with or without microtubule network disruption by Nocodazole (Noco, 13 μM) for 4 hr prior to insulin. The dynamics of GLUT4-GFP in the different conditions are also exemplified in *Figure 2—video 1*. (**B**) Quantified microtubule-based GLUT4 trafficking. (**C**) Representative images of muscle fibers ± pre-treatment with Noco 13 μM, for 15 hr, followed

*Figure 2 continued on next page*

*Figure 2 continued*

by recovery in Noco-free medium for 9 hr. (**D**) Quantification of GLUT4 distribution between the microtubule nucleation sites (structures sized >4 µm²), intermediate-sized structures (0.4–4 µm²) and the smallest resolvable structures (<0.4 µm²) in fibers treated as in C. Compartment identification is described in *Figure 2—figure supplement 1E*. (**E**) GLUT4 and Syntaxin6 (Stx6) in muscle fiber from human vastus lateralis muscle. (**F**) GLUT4-Stx6 overlap in perinuclear region of mouse flexor digitorum brevis muscle fibers in Dimethylsulfoxide (DMSO) medium with and without Noco (13 µM) treatment. For A, B, $n \geq 14$ muscle fibers from 5 different mice. For C, D, $n = 9$–11 muscle fibers from 3 different mice. For E, $n = 3$ subjects. Data are presented as mean with individual data points. \*\*\*$p < 0.001$ different from basal, ###$p < 0.001$ different from INS, ##$p < 0.01$ different from Noco recovery. ¤¤¤$p < 0.001$ analysis of variance (ANOVA) effect. Scale bar = 5 µm (**A–C**) and 2 µm (**E**).

The online version of this article includes the following video, source data, and figure supplement(s) for figure 2:

**Source data 1.** Data used for quantification of GLUT4-Stx6 overlap in perinuclear region of mouse flexor digitorum brevis muscle fibers in DMSO medium with and without Noco (13 µM) treatment.

**Figure supplement 1.** Glucose transporter 4 (GLUT4) trafficking and localization was dependent on an intact microtubule network.

**Figure supplement 1—source data 1.** Data used for quantification of *Figure 2—figure supplement 1A–D, G* and raw unedited blots for *Figure 2—figure supplement 1A*.

**Figure 2—video 1.** Glucose transporter 4 (GLUT4) movement was disrupted by Nocodazole treatment.

https://elifesciences.org/articles/83338/figures#fig2video1

---

(*Figure 2E, F*, *Figure 2—figure supplement 1H*). Altogether, our data demonstrate that GLUT4 trafficking and distribution is disrupted by pharmacological microtubule network depolymerization in a fully reversible manner.

## Prolonged, but not short-term, microtubule disruption blocked insulin-induced muscle glucose uptake

To investigate the requirement of microtubule-based GLUT4 trafficking and localization for insulin-induced muscle glucose uptake, we assessed muscle glucose uptake ± insulin and ± microtubule disruption in isolated incubated intact mouse soleus and extensor digitorum longus (EDL) muscles. When mouse soleus and EDL muscles were incubated ex vivo ± insulin and ± Noco (13 µM) for 15 min and up to 2 hr, an interaction between insulin and Noco was observed and the insulin-induced glucose uptake was gradually impaired over time and completely disrupted after 2 hr in both muscles (*Figure 3A*). The increase by insulin stimulation was significantly impaired after 40 min in soleus and 2 hr in EDL muscle (*Figure 3B*). Insulin-stimulated phospho-signaling via Akt and TBC1D4 was unaffected by Noco treatment (*Figure 3—figure supplement 1A–C*).

To understand the temporal resolution of microtubule network disruption and its effect on insulin-induced glucose uptake, we investigated glucose uptake adult isolated single fibers in real time using a custom-made perifused organ-on-chip system (*Gowers et al., 2015*; *Trouillon and Gijs, 2016*) featuring a glucose-sensing electrode for glucose uptake measurements (*Figure 3C*). In brief, this chip continuously measures glucose concentration in perifusate post muscle fiber exposure, allowing the estimation of glucose uptake over time. We confirmed the ability of the chip to measure glucose uptake in skeletal muscle fibers (*Figure 3—figure supplement 1D–F*). Specifically, the chip measured glucose uptake in isolated FDB fibers at ~5 µM glucose concentration sensitivity (defined as a registered fluctuation of thrice the standard deviation [SD] of the baseline measurements) and a temporal resolution of <4 s (*Figure 3—figure supplement 1G, H*). Noco (13 µM) caused complete microtubule disruption in FDB fibers within 5 min similar to a 2-hr treatment (*Figure 3D*). Interestingly, acute microtubule disruption (5 min Noco) affected neither basal (*Figure 3—figure supplement 1I*) nor insulin-induced muscle glucose uptake, whereas 2 hr treatment by Noco or colchicine (25 µM), another microtubule network disrupter, completely blocked insulin-induced muscle glucose uptake (*Figure 3E, F*). Notably, GLUT4-containing large membrane structures corresponding mainly to microtubule nucleation sites were already reduced after 5 min of Noco exposure, whereas accumulation of GLUT4 in intermediate- and small-sized membrane structures was only observed after 2 hr of Noco exposure (*Figure 3G, H*). Thus, an intact microtubule network is not required for the immediate insulin-induced GLUT4 translocation response in adult skeletal muscle fibers. However, prolonged disruption of the microtubule network causes a more pronounced missorting of GLUT4 and renders skeletal muscle unresponsive to insulin.

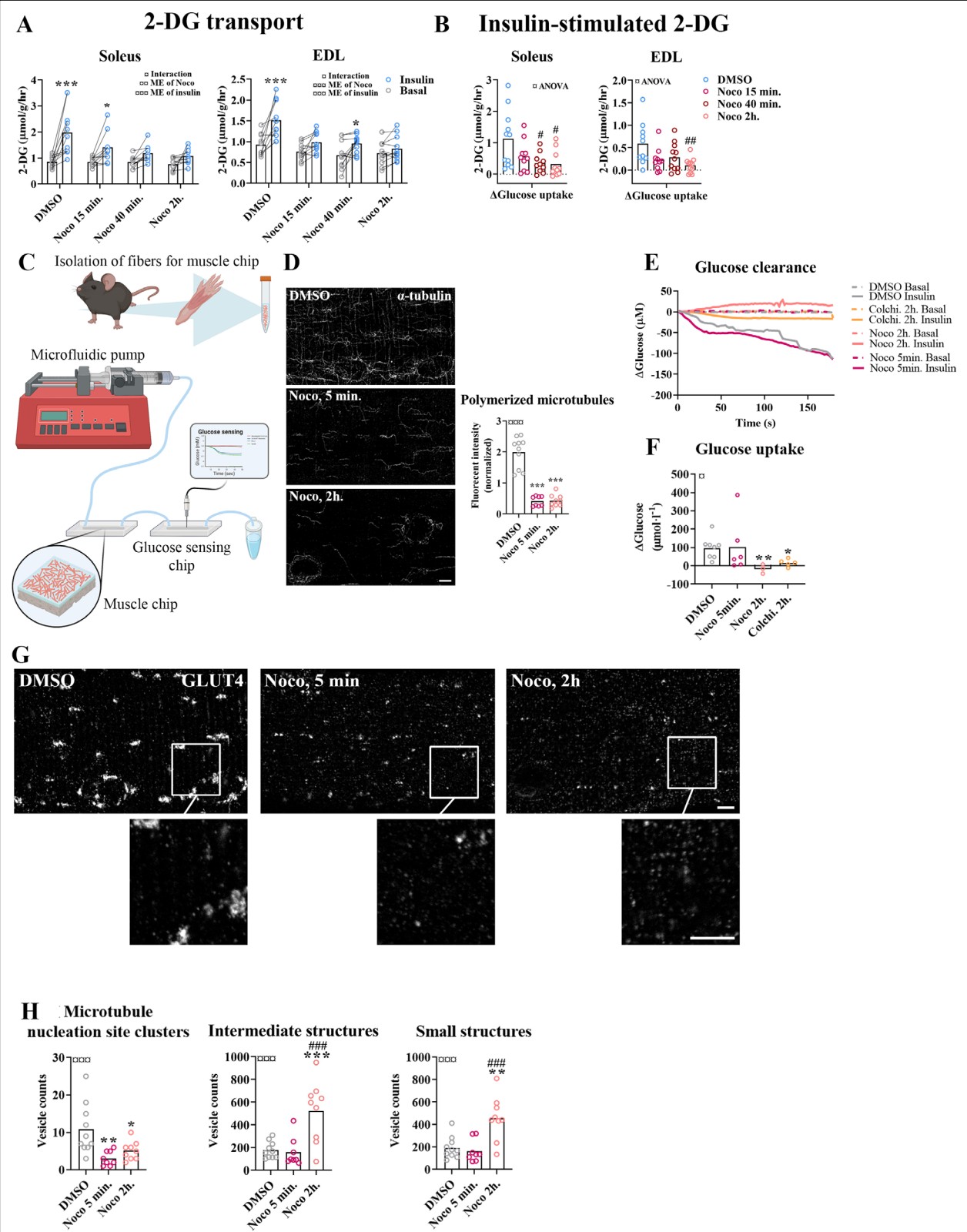

**Figure 3.** Time-dependent effect of microtubule disruption on insulin-induced muscle glucose uptake. (**A**) 2-Deoxyglucose (2-DG) transport in basal and insulin-stimulated mouse soleus and extensor digitorum longus (EDL) muscles pretreated with Nocodazole (Noco, 13 µM) for the indicated time. (**B**) Insulin-stimulated 2-DG transport (insulin minus basal) from muscles shown in A. (**C**) Experimental setup for muscle-on-a-chip system with glucose sensor. (**D**) Microtubules imaged with α-tubulin in glucose transporter 4 (GLUT4)-GFP-expressing mouse flexor digitorum brevis (FDB) fibers treated ±

*Figure 3 continued on next page*

*Figure 3 continued*

Noco (13 µM) for 5 min or 2 hr. (**E**) 180-s measurements of glucose concentration in perifusate from basal and insulin-treated FDB muscle fibers in muscle chips pre-incubated with DMSO, Noco (13 µM, 5 min or 2 hr) or colchicine (25 µM, 2 hr). (**F**) Insulin-stimulated glucose uptake into FDB muscle fibers in muscle chips calculated from the last 20 s of the concentration curves in E. (**G**) Representative GLUT4 images from isolated mouse FDB muscle fibers treated ± Noco (13 µM) for 5 min or 2 hr. (**H**) Quantification of GLUT4 in large, intermediate- and small-sized membrane structures in FDB fibers treated ± Noco (13 µM) for 5 min or 2 hr. The membrane compartment division by size is shown in *Figure 2E*. For A, B, *n* = 6–7 muscles from 6 to 7 mice. For **D, G, H**, *n* = 8–10 muscle fibers from 3 different mice. For **E, F**, *n* ≥ 3 muscle chips from 3 to 4 mice. Data are presented as mean with individual data points. Paired observations from the same mouse are indicated by a connecting line. */**/***$p < 0.05/0.01/0.001$ different from basal/DMSO, #/##/###$p < 0.05/0.01/0.001$ different from DMSO. ¤/¤¤/¤¤¤$p < 0.05/0.01/0.001$ analysis of variance (ANOVA) effect. Scale bar = 5 µm.

The online version of this article includes the following source data and figure supplement(s) for figure 3:

**Source data 1.** Data used for quantification of 2-DG transport and glucose clearance and uptake in *Figure 3A, B, E, F*, polymerized microtubules in *Figure 3D* and glucose transporter 4 (GLUT4) localization in *Figure 3H*.

**Figure supplement 1.** Time-dependent effect of microtubule disruption on insulin-induced muscle glucose uptake.

**Figure supplement 1—source data 1.** Data used for quantification of protein expression and electrochemical glucose sensing in *Figure 3—figure supplement 1*.

## KIF5B-containing kinesin-1 motor proteins regulate muscle GLUT4 trafficking

Next we investigated which motor protein(s) mediate microtubule-dependent GLUT4 trafficking in skeletal muscle. The kinesin-1 heavy chain protein KIF5B has been implicated in GLUT4 trafficking in adipocytes (*Semiz et al., 2003*; *Habtemichael et al., 2018*). Thus, we studied the effect of kinesore, a small molecule modulator which both inhibits kinesin-1 interaction with specific cargo adaptors but also stimulates Kinesin motor function (*Randall et al., 2017*), in incubated soleus and EDL muscles and differentiated primary human myotubes as well as the effect of *Kif5b* short hairpin (sh) RNA knockdown and kinesore in L6 skeletal muscle cells overexpressing exofacially tagged GLUT4 (*Kishi et al., 1998*; *Wang et al., 1998*; *Figure 4A*). In both soleus and EDL muscle, 2 hr of kinesore (50 µM) exposure strongly impaired insulin-stimulated glucose uptake (*Figure 4B*) without affecting p-Akt Ser473 and slightly increased basal and insulin-stimulated p-TBC1D4 Thr642 (*Figure 4—figure supplement 1A*). AMPK is an insulin-independent stimulator of GLUT4 translocation which may indirectly stimulate TBC1D4 Thr642 phosphorylation (*Kjøbsted et al., 2015*). Notably, phosphorylation of AMPK and downstream ACC2 were stimulated by kinesore in both basal and insulin-stimulated soleus and EDL muscles (*Figure 4—figure supplement 1B*). In primary human myotubes differentiated for 7 days kinesore (50 µM) and Noco (13 µM) reduced the glucose uptake and completely blocked the insulin response (*Figure 4C*). In L6 myoblasts, we lowered KIF5B expression using shRNA by ~70% in L6 myoblasts (*Figure 4—figure supplement 1C*). This did not affect GLUT4 expression (*Figure 4—figure supplement 1D*) but impaired insulin-stimulated GLUT4 translocation (*Figure 4D*). Unlike the inhibitory effect in incubated mouse muscle and primary human myotubes, kinesore-stimulated GLUT4 translocation and glucose uptake additively to insulin in L6 muscle cells, and modestly potentiated insulin-stimulated Akt Thr308 without affecting AMPK signaling (*Figure 4—figure supplement 1E–J*). Collectively, although kinesore surprisingly had a stimulatory and seemingly MT-independent effect on GLUT4 translocation in L6 muscle cells, our shRNA data in L6 myoblasts and kinesore data in adult muscle and primary human myotubes support the requirement of KIF5B-containing kinesin-1 motor proteins in GLUT4 translocation in skeletal muscle.

## Insulin resistance induced by C2 ceramide and high-fat diet impaired microtubule-based GLUT4 trafficking

Having established an essential role for the microtubule network in GLUT4 trafficking and muscle glucose uptake, we proceeded to test if microtubule-based GLUT4 trafficking was impaired in insulin-resistant states. We induced insulin resistance in adult mouse skeletal muscle both in vitro using short-chain C2 ceramide and in vivo using diet-induced obesity (*Figure 5A*). In vitro, treatment of isolated FDB muscle fibers with C2 ceramide (50 µM) impaired Akt Thr308 phosphorylation (*Figure 5—figure supplement 1A*) and in insulin-stimulated fibers markedly reduced microtubule-based GLUT4 trafficking defined as the number of moving GLUT4 structures (*Figure 5B*) and the total microtubule-based traveling of GLUT4 structures (*Figure 5—figure supplement 1B*). In vivo, mice

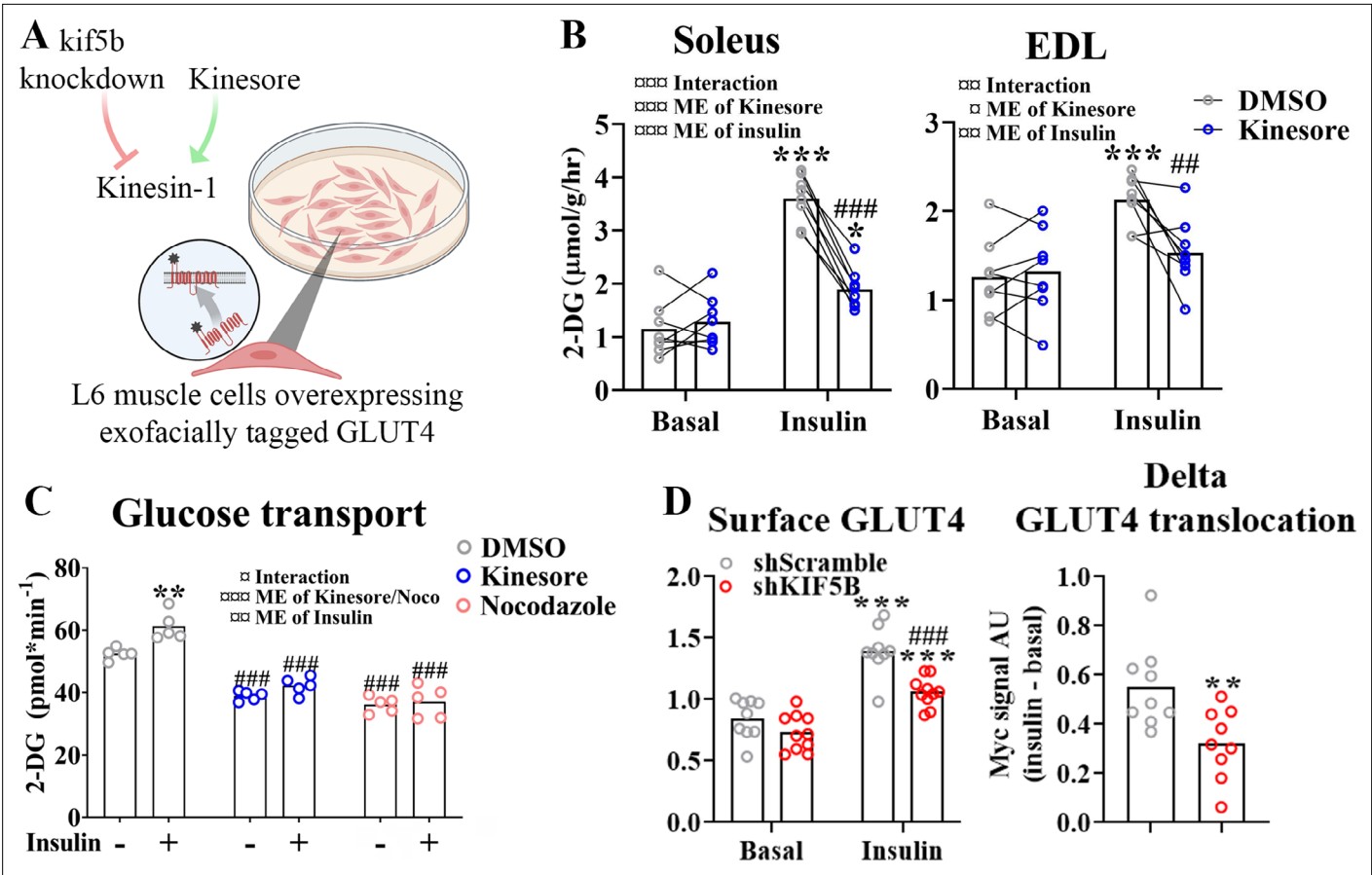

**Figure 4.** Kinesin-1 containing KIF5B-regulated glucose transporter 4 (GLUT4) localization and translocation. (**A**) Schematic overview of L6 muscle cell system to assess GLUT4 surface content. (**B**) 2-Deoxyglucose (2-DG) transport in basal and insulin-stimulated mouse soleus and extensor digitorum longus (EDL) muscles pretreated with kinesore (50 μM) for 2 hr. (**C**) Deoxyglucose (2-DG) transport in basal and insulin-stimulated primary human myotubes pretreated ± kinesore (50 μM) or Noco (13 μM) for 2 hr. (**D**) Exofacial GLUT4 signal in serum starved (4 hr) basal and insulin-stimulated (100 nM, 15 min) L6 myoblasts (left) and insulin response (insulin minus basal, right) in GLUT4 surface content. L6 myoblasts were transfected with short hairpin scramble RNA (shScramble) or shRNA targeting *Kif5b* 72 hr prior to the experiment. Analysis of variance (ANOVA) main effect of insulin (¤¤¤) and *shKif5b* (¤¤¤) and interaction (¤). (**B**) *n* = 8 muscles in each group, lines indicate muscles from same mouse. (**C**) Each data point represents the average of 3 replicates and originate from at least 3 independent experiments. Data are presented as mean with individual data points. */**/***p < 0.01/0.001 effect of insulin. ##/###p < 0.01/0.001 different from DMSO/Scramble. ¤/¤¤/¤¤¤p < 0.05/0.01/0.001 ANOVA effect.

The online version of this article includes the following source data and figure supplement(s) for figure 4:

**Source data 1.** Data used for quantification of glucose transporter 4 (GLUT4) localization and GLUT4 surface content in *Figure 4*.

**Figure supplement 1.** Kinesin-1 containing KIF5B-regulated glucose transporter 4 (GLUT4) localization and translocation.

**Figure supplement 1—source data 1.** Data used for quantification of glucose transporter 4 (GLUT4) surface content, protein expression, and glucose uptake in *Figure 4—figure supplement 1*.

fed a 60% high-fat diet (HFD) for 10 weeks exhibited impaired tolerance to insulin and glucose as well as reduced insulin-stimulated phosphorylation of Akt Thr308 and Akt substrate TBC1D4 Thr642 in isolated FDB fibers (*Figure 5—figure supplement 1C–E*), confirming whole-body and skeletal muscle insulin resistance. Similar to C2 ceramide-treated fibers, HFD-exposed FDB muscle fibers exhibited impaired microtubule-based GLUT4 trafficking (*Figure 5C*, *Figure 5—figure supplement 1F*). This prompted us to investigate whether the microtubule polymerization was itself insulin responsive and/or affected by insulin resistance. To test this, we transfected mouse FDB muscle fibers with the microtubule plus-end-binding protein EB3-GFP, which binds the tip of growing microtubules via its calponin homology domain and has previously been used for live-cell characterization of microtubule polymerization (*Stepanova et al., 2003*). As previously reported (*Oddoux et al., 2013*), EB3-GFP transfection allowed visualization of growing microtubules as a dynamic comet tail-like appearance,

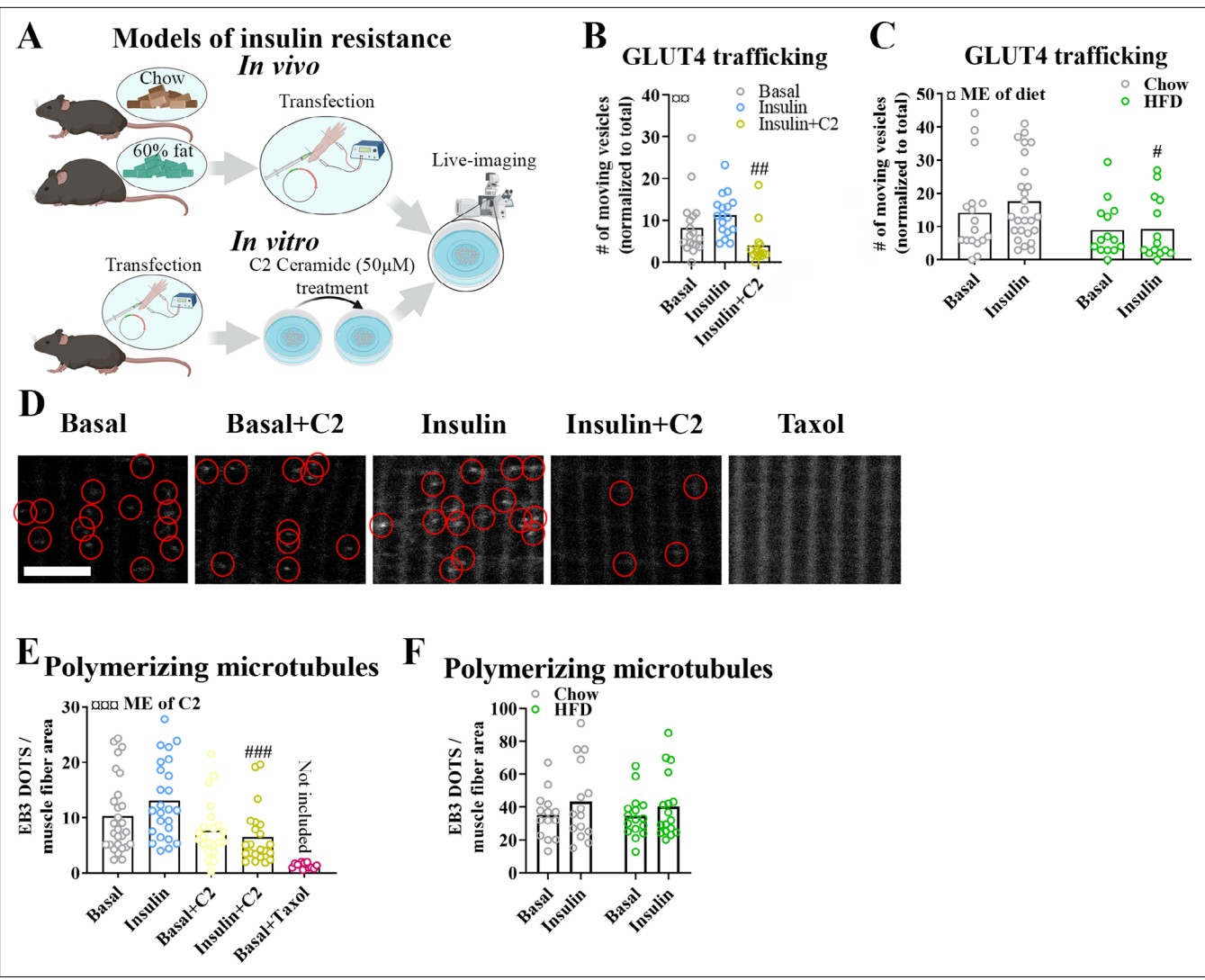

**Figure 5.** Insulin resistance impairs microtubule-based glucose transporter 4 (GLUT4) trafficking. (**A**) Overview of in vitro and in vivo insulin resistance models used. (**B**) Quantified microtubule-based GLUT4 trafficking in basal, insulin (INS, 30 nM) and insulin + C2 ceramide (C2) (INS + C2, 30 nM + 50 μM) treated flexor digitorum brevis (FDB) muscle fibers. (**C**) Quantified microtubule-based GLUT4 trafficking in basal or INS (30 nM) treated FDB fibers from chow or high-fat diet (HFD) fed mice. (**D**) Representative images of polymerizing microtubules in EB3-GFP-expressing FDB muscle fibers treated ± C2 (50 μM), paclitaxel (Taxol, 10 μM) for 2 hr prior to 15–30 min of INS (30 nM) stimulation. Red circles highlight microtubule tip-bound EB3-GFP. (**E**) Quantification of polymerizing microtubules based on EB3-GFP in FDB fibers treated as in D. (**F**) Quantification of polymerizing microtubules based on EB3-GFP in FDB fibers isolated from chow or 60% HFD fed mice and treated ± INS (30 nM) for 15–30 min. For **B–F**, n ≥ 13 muscle fibers from 3 to 4 mice. Taxol-treated muscle fibers were only used as a control and not included in the statistical analysis. NA = not statistically analysed. Data are presented as mean with individual data points. #/##/###p < 0.05/0.01/0.001 different from INS (**B**) or different from corresponding group in chow fed mice (**C**) or control fibers (**E**). □/□□/□□□p < 0.05/0.01/0.001 main effect (ME) of diet/C2.

The online version of this article includes the following video, source data, and figure supplement(s) for figure 5:

**Source data 1.** Data used for quantification of glucose transporter 4 (GLUT4) trafficking and microtubule polymerization in *Figure 5B, C, E, F*.

**Figure supplement 1.** Insulin resistance impairs microtubule-based glucose transporter 4 (GLUT4) trafficking.

**Figure supplement 1—source data 1.** Data used for quantification of protein expression, glucose transporter 4 (GLUT4) trafficking, glucose and insulin tolerance, microtubule polymerization and polymerization directionality in *Figure 5—figure supplement 1*.

**Figure 5—video 1.** Polymerizing microtubules in muscle fibers.

https://elifesciences.org/articles/83338/figures#fig5video1

an effect completely prevented by the microtubule stabilizer taxol (10 μM) (*Figure 5—figure supplement 1G*, *Figure 5—video 1*). In our datasets, we analyzed the microtubule polymerization frequency (by counting EB3-GFP puncta; *Komarova et al., 2009*), the average polymerization distance, the total polymerization distance and the polymerization directionality following C2 ceramide exposure or HFD. Insulin tended (p = 0.095) to increase the number of polymerizing microtubules by an average of 28% compared to basal fibers while C2 ceramide treatment reduced the amount of polymerizing microtubules significantly and taxol almost abolished microtubule polymerization (*Figure 5D, E*). C2 ceramide treatment also reduced the total polymerization distance while the average polymerization was unaffected (*Figure 5—figure supplement 1H*).

In contrast to the effect of C2 ceramide on microtubule dynamics, HFD-induced insulin resistance was not associated with alterations in the number of polymerizing microtubules, average polymerization distance or total polymerization distance (*Figure 5F*, *Figure 5—figure supplement 1I*). The polymerization directionality was also not affected by HFD (*Figure 5—figure supplement 1J*). Altogether, different models of insulin resistance impaired microtubule-based GLUT4 trafficking in adult muscle fibers, suggesting a role in adult skeletal muscle insulin resistance. In contrast, defective microtubule polymerization was observed with C2 ceramide but not with the presumably more physiologically relevant HFD insulin resistance model.

## Discussion

In the present study, we provide translational evidence in adult human and mouse skeletal muscle, showing that the microtubule network is crucial for long-range directional GLUT4 trafficking via motor proteins, likely including KIF5B. Microtubule polymerization in isolated mouse muscle fibers could be abolished pharmacologically within minutes without affecting insulin-stimulated glucose uptake whereas longer pharmacological inhibition progressively caused GLUT4 mislocalization and lowered insulin responsiveness. These data are consistent with a model where microtubules are required for correct intramyocellular GLUT4 compartmentalization but not the ultimate insulin-stimulated GLUT4 translocation to the cell surface from these compartments. Importantly, microtubule-based GLUT4 movement was impaired in two classical mouse insulin resistance models, short-chain ceramide treatment and diet-induced obesity. These data implicate dysregulation of microtubule-mediated GLUT4 trafficking and their localization in the etiology of adult skeletal muscle insulin resistance.

What may cause the reduced number of GLUT4 moving on microtubules in insulin-resistant muscle? Based on cell-culture studies, GLUT4 is packaged into specialized GSVs, which upon insulin stimulation can undergo exocytosis (*Fazakerley et al., 2022*). This exocytosis may include an intermediate step involving release and heterotypic fusion of GSVs with transferrin receptor positive endosomes observable using super-resolution quantum dot single GLUT4 particle tracking in isolated mouse soleus and EDL muscle fibers (*Hatakeyama and Kanzaki, 2017*). Such a model is consistent with previous reports of insulin-stimulated release of GSVs into an endosomal recycling pool in 3T3-L1 adipocytes (*Coster et al., 2004*; *Govers et al., 2004*). In 3T3-L1 adipocytes, GSVs are synthesized via the trans-Golgi network and the Endoplasmic reticulum (ER)–Golgi intermediate compartment (*Gould et al., 2020*) by mechanisms involving TBC1D4 (*Bruno et al., 2016*; *Brumfield et al., 2021*; *Sano et al., 2007*). We note that TBC1D4 phosphorylation on multiple sites is often decreased in insulin-resistant skeletal muscle of humans (*Pehmøller et al., 2012*; *Vind et al., 2011*) and rodents (*Castorena et al., 2014*; *Li et al., 2019*). Conversely, TBC1D4 phosphorylation is increased after insulin-sensitizing exercise/contraction (*Pehmøller et al., 2012*; *Funai et al., 2009*; *Arias et al., 2007*; *Treebak et al., 2009*; *Kjøbsted et al., 2019*; *Kjøbsted et al., 2017*) which might augment the insulin-responsive GLUT4 pool (*Geiger et al., 2006*; *Knudsen et al., 2020c*; *Bradley et al., 2014*). The presently observed decreased GLUT4 movement on microtubules could reflect impaired availability of GLUT4 for trafficking, i.e. impaired TBC1D4-dependent GSV biosynthesis and/or heterotypic fusion of GSVs with transferrin receptor positive endosomes. Another possibility is that recruitment of GLUT4 onto microtubules, i.e. release of tethered GSVs and/or motor protein interaction, is impaired. In 3T3-L1 adipocytes, insulin stimulated the frequency but not movement speed of GLUT4 on microtubules, an effect blocked by a dominant-negative cargo-binding kinesin light chain 1 (KLC1) which impairs KIF5B function (*Semiz et al., 2003*). Also, KLC1 was proposed to regulate GLUT4 translocation in 3T3-L1 adipocytes (*Semiz et al., 2003*) and L6 myoblasts via insulin-regulated interaction with Double C2 domain B (*Zhang et al., 2019*). These data are consistent with our observations of KIF5B-dependent GLUT4

translocation in insulin-stimulated L6 muscle cells. KIF5B was proposed to regulate GSV trafficking via insulin-stimulated binding to TUGUL, a cleavage product of the intracellular GSV tethering protein TUG (*Habtemichael et al., 2018*; *Bogan et al., 2003*). Multiple studies by Jonathan Bogan's group linked TUG to skeletal muscle insulin-stimulated GLUT4 traffic and glucose uptake (*Belman et al., 2015*; *Löffler et al., 2013*). Insulin-stimulated TUG cleavage and expression of the putative protease Usp25m were reduced in rat adipose tissue after a 3-day high-fat/high-sugar diet-induced insulin resistance (*Habtemichael et al., 2018*), suggesting possible dysregulation in insulin resistance. Apart from KIF5B and TUG, GLUT4 was proposed to utilize other motor proteins to move on microtubules in 3T3-L1 adipocytes, i.e. dynein via Rab5 (*Huang et al., 2001*), and the kinesin-2 family member KIF3A via AXIN and the ADP-ribosylase tankyrase 2 (*Guo et al., 2012*; *Imamura et al., 2003*). Whether any of these mechanisms regulate GLUT4 traffic in skeletal muscle should be tested further.

Noco treatment potently reduced the number of large GLUT4-containing structures, while causing the number of medium-sized GLUT4-containing structures to increase within 2 hr of treatment. We speculate whether the medium-sized GLUT4 structures are so-called Golgi mini-stacks, small immature membrane compartments of Golgi origin incapable of protein secretion, reported to accumulate in microtubule disrupted cells (*Fourriere et al., 2016*). Alternatively, the membrane structures accumulated in Noco-treated fibers might be internalized GLUT4 in endosomal structures unable to reach microtubule nucleation sites via retrograde transport. Future studies should characterize the nature of these membrane structures by e.g. measuring different compartment-specific protein markers.

Our glucose uptake time-course data in perifused mouse FDB fibers suggest that microtubules are required to build and maintain the pool of insulin-responsive GLUT4 vesicles near the cell surface observed in previous studies (*Lizunov et al., 2012*), and that disruption of this process may contribute to insulin-resistant GLUT4 translocation in adult skeletal muscle. If true, then one would predict GLUT4 mislocalization to be observable in unstimulated insulin-resistant skeletal muscle. Consistent with this prediction, Garvey et al. measured GLUT4 content in different sucrose density fractions in human vastus lateralis muscle biopsies and found a basal redistribution of GLUT4 to a denser fraction in type 2 diabetic subjects compared to control (*Garvey et al., 1998*). Similar observations of altered intracellular GLUT4 distribution were made using a different fractionation protocol in subcutaneous adipose biopsies from type 2 diabetic subjects compared to control (*Maianu et al., 2001*), implying that GLUT4 mislocalization is a shared hallmark of insulin-resistant muscle and adipose cells. Application of higher-resolution microscopy to the study of basal and insulin-stimulated GLUT4 localization in adult human and rodent insulin-resistant skeletal muscle could help to resolve the relative distribution of GLUT4 between specific membrane compartments.

A previous study reported that the microtubule disrupting drug colchicine (25 μM for up to 8 hr) had no effect on insulin and contraction-stimulated glucose uptake, whereas Noco (3–83 μM for 30–60 min) potently inhibited glucose uptake in ex vivo incubated adult rat muscle and 33 μM Noco added for <5 min directly inhibited GLUT4 transporter activity into sarcolemma derived giant vesicles (*Huang et al., 2001*). Regarding colchicine, these divergent findings mirror previous studies in 3T3-L1 adipocytes where some support the absent effect of colchicine on insulin-stimulated cell surface GLUT4 translocation/glucose uptake despite strong microtubule disruption (*Molero et al., 2001*; *Huang et al., 2001*) whereas others found colchicine to inhibit insulin-stimulated GLUT4 translocation (*Fletcher et al., 2000*; *Emoto et al., 2001*). The reason for this variation between studies is not readily apparent since the models and colchicine treatment protocols overlap. Regarding Noco, a direct inhibitory effect of 33 μM Noco added for >2 min on GLUT4 activity was suggested by previous 3T3-L1 adipocyte studies (*Molero et al., 2001*; *Shigematsu et al., 2002*; *Huang et al., 2005*), However, Molero et al. also observed no inhibition of insulin-stimulated glucose uptake using 2 μM Noco for 1 hr despite complete disruption of microtubules (*Molero et al., 2001*), suggesting that Noco is useful to study the microtubule-glucose uptake connectivity at low concentrations. Since Noco was shown to inhibit GLUT4 activity within 2 min and we observed no effect of 13 μM Noco in isolated muscle fibers on basal glucose uptake or insulin-stimulated glucose uptake after 5 min at a concentration causing maximal inhibition of microtubule polymerization, we find it unlikely that Noco at the concentration used had major effects on GLUT4 activity.

We were unable to detect a significant effect of insulin on the number of GLUT4 moving on microtubules in the individual experiments performed in the current study. This contrasts the previously reported observation that rodent skeletal muscle expressing GFP-tagged GLUT4 had increased

recovery of fluorescence after photo bleaching when stimulated by insulin, suggesting that insulin increased the overall GLUT4 movement (*Lauritzen et al., 2008*). Meanwhile, in primary and 3T3-L1 adipocyte cell culture total internal reflection fluorescence (TIRF) microscopy studies, insulin increased the number of GLUT4 halting and docking beneath the plasma membrane prior to insulin-stimulated insertion (*Lizunov et al., 2005*; *Gonzalez et al., 2006*; *Lizunov et al., 2009*; *Bai et al., 2007*). TIRF imaging of mouse muscle fibers expressing HA-GLUT-GFP suggested a similar GLUT4 halting and fusion-promoting effect of insulin in skeletal muscle (*Lizunov et al., 2012*). Regarding microtubule dynamics, previous studies reported that insulin increased microtubule polymerization and/or density in 3T3-L1 adipocytes (*Parker et al., 2019*; *Olson et al., 2003*; *Dawicki-McKenna et al., 2012*) and in L6 skeletal muscle cells (*Liu et al., 2013*) but insulin also decelerated CLASP2-positive MT polymerization in 3T3-L1 adipocytes (*Parker et al., 2019*). In adult muscle, we did not detect any significant effect of insulin on the assessed parameters. Given that insulin may differentially increase or decrease the mobility of subpopulations of GLUT4 and microtubules, it seems likely that our relatively crude analyses of total subsarcolemmal GLUT4 movement could mask larger or even opposite behaviors of specific GLUT4 subpopulations.

The actin cytoskeleton is known to exhibit extensive interactions with microtubules (*Pimm and Henty-Ridilla, 2021*). Importantly, the actin cytoskeleton and associated regulators and binding proteins were suggested in L6 muscle cells and adult mouse muscle to be required for skeletal muscle GLUT4-dependent glucose transport (*Sylow et al., 2013a*; *Sylow et al., 2013b*; *JeBailey et al., 2007*; *Tong et al., 2001*; *Brozinick et al., 2004*; *Ueda et al., 2010*; *Toyoda et al., 2011*; *Kee et al., 2015*), likely by promoting the final steps of GLUT4 docking and fusion (*Zaid et al., 2008*). In adult muscle, the cytoplasmic beta and gamma-actin isoforms are lowly expressed compared to cultured cells and their individual KO in mouse muscle does not affect insulin-stimulated glucose uptake, making the exact actin isoform and mechanistic involvement of actin unclear (*Madsen et al., 2018*). Nevertheless, multiple studies support a role of actin in regulating adult muscle GLUT4-dependent glucose uptake. Hence, the interconnectivity between the actin and microtubule cytoskeletons in mediating GLUT4 trafficking should be investigated.

In conclusion, we presently demonstrated that GLUT4 is present on microtubules in adult mouse and human skeletal muscle and that acute microtubule disruption causes intramyocellular GLUT4 redistribution and eventually decreases insulin responsiveness of glucose transport. Decreased microtubule-dependent GLUT4 movement was observed in in vitro and in vivo mouse insulin resistance models, suggesting that disturbed microtubule-based GLUT4 trafficking is a feature of insulin resistance in adult skeletal muscle.

## Materials and methods
### Sample obtaining
*Human muscle samples* are tissue from *m. vastus lateralis* from young healthy men fasted for 6–7 hr. Further details on the subjects and tissue processing are described in previously published studies (*Knudsen et al., 2020c*; *Steenberg et al., 2020*). The study was approved by the Regional Ethics Committee for Copenhagen (H-6-2014-038; Copenhagen, Denmark) and complied with the guidelines of the 2013 Declaration of Helsinki. Written informed consent was obtained from all participants prior to entering the study.

*Mouse muscle samples* were from 10- to 16-week-old C57BL/6 mice. All animal experiments were approved by the Danish Animal Experimental Inspectorate or by the local animal experimentation committee of the Canton de Vaud under license 2890 and complied with the European Union legislation as outlined by the European Directive 2010/63/EU. The current work adheres to the standards outlined in the ARRIVE reporting guidelines. Male C57BL/6NTac mice, 16 weeks old, were used for the experiments including HFD fed mice. The mice were fed a 60% HFD or a control standard rodent chow diet ad libitum. For the rest of the experiments, the mice were female C57BL/6JRj aged 10–16 weeks fed ad libitum with a standard rodent chow diet. All mice were exposed to a 12 hr:12 hr light–dark cycle.

### Gene transfer and fiber isolation
FDB muscles were electroporated in vivo similar to *DiFranco et al., 2009* and isolated as previously described (*Knudsen et al., 2019*). The following plasmids were used: pB-GLUT4-7myc-GFP plasmid

(a gift from Jonathan Bogan, Addgene plasmid #52872); p-mCherry-Tubulin-C1 (a gift from Kristien Zaal); HA-GLUT4-EOS (originally from the Zimmerberg laboratory, *Lizunov et al., 2013*, was a gift from Timothy McGraw); and p-EB3-GFP-N1 (originally from the Akhmanova laboratory, *Stepanova et al., 2003*, was a gift from Evelyn Ralston).

## Fiber culturing and drug treatments

Experiments with isolated fibers were performed the day after isolation. For prolonged (15 hr) nocodazole treatment, nocodazole (M1404, Merck) was added for a final concentration of 4 µg/ml at this step. When palmitic acid treatment is indicated, this was added for a final concentration of 0.5 mM at this step as well. Palmitic acid was dissolved to a 200 mM solution in 1:1 ethanol and α-minimal essential medium (MEM), from which a 16× solution containing 100 mg/ml fatty acid-free bovine serum albumin (BSA) was made. Non-treated fibers were treated with BSA without palmitic acid. When indicated C2 ceramide (50 µM) (860502, Avanti), Paclitaxel (10 µM) (T7402, Merck), kinesore (6664, Tocris), or colchicine (25 µM) (C9754, Sigma) was added 2 hr prior to imaging/lysing whereas nocodazole (13 µM) was added 4 hr prior unless otherwise mentioned. For signaling analyses, 30 nM insulin (Actrapid, Novo Nordisk A/S) was added 15 min prior to lysing, for microscopic analyses 30 nM insulin was added 15–30 min prior to imaging. For fixation fibers were incubated with 4% paraformaldehyde (Electron Microscopy Sciences) in phosphate-buffered saline (PBS) for 20 min.

## Cell culturing and experiments

Mycoplasma-free L6 rat myoblasts expressing myc-tagged GLUT4 (*Wang et al., 1998*) were maintained in α-MEM (12561056, Gibco) supplemented with 10% fetal bovine serum and 1% pen/strep antibiotic in a humidified atmosphere containing 5% $CO_2$ and 95% air at 37°C. Differentiation to myotubes were achieved by lowering the serum concentration to 2% for 7 days. Visual myotube formation was used as authentication. For specific knockdown of KIF5B, shRNA constructs containing a 19-nucleotide (GGACAGATGAAGTATAAAT) sequence derived from mouse *Kif5b* mRNA (*Zhao et al., 2020*; a gift from Dr. Kwok-On Lai, City University of Hong Kong) were used using JetPRIME (Polyplus) according to the manufacturer's protocol. As control shRNA with the sequence CCTAAGGTTAAGTCGCCCTC GCTCGAGCGAGGGCGACTTAACCTTAGG (Addgene plasmid # 1864) were used. Three days after initial transfection, the experiments were conducted as described in the figure legends. For GLUT4 translocation assessment, cell surface GLUT4myc was detected using a colorimetric assay (*Wijesekara et al., 2006*). Drug treatments were performed as described in the figure legends.

Human primary myoblasts (SK111, Cook MyoSite) were maintained and passaged in myotonic basal media (MB-2222, Cook MyoSite) supplemented with 10% myotonic basal media (MB-3333, Cook MyoSite) and 1% P/S antibiotic in a humidified atmosphere containing 5% $CO_2$ and 95% air at 37°C. For differentiation and 2-DG transport measurements myoblasts were seeded at 50,000 cells/cm$^2$ in collagen coated 96-well Cytostar-T scintillation plates (Perkin Elmer). After 2 days differentiation was initiated by switching to MEM (Gibco, 41090-028) with 2% horse serum, 10 µM 1-Dimethylethyl Ester (DAPT, Stemcell Technologies) and 1 µM Dabrafenib (Stemcell Technologies). Visual myotube formation was used as authentication. After 7 days of differentiation, myotubes were starved in PBS with magnesium and calcium and 10% MEM in 3.5 hr with DMSO/kinesore/Noco included the last 2 hr. Next, cells were incubated 15 min ± insulin (100 nM) before $^{14}$C-labeled 2-DG were added and tracer accumulation were measured for 3 hr.

## Glucose uptake measurements

*2-DG transport* into Soleus and EDL muscles were assessed as described before (*Knudsen et al., 2020b*).

*2-DG transport* into L6 cells was measured by washing cells in PBS containing 4-(2-hydroxyethyl)-1-piperazineethanesulfonic acid (HEPES) and incubating them in PBS + HEPES containing 2-[3*H*] deoxyglucose for 5 min before cell harvest in lysis buffer. Tracer accumulation was then measured by liquid scintillation counting.

*Electrochemical glucose sensing* – Inspired by *Trouillon et al., 2017*, we fabricated a microfluidic chip system using standard soft lithographic techniques. The chip was divided into a tissue chamber and a glucose-sensing chamber connected by tubing. Both chambers were molded based on SU-8 master wafers. The tissue chamber consisted of two identical units each containing three layers of

**Figure 6.** Overview of muscle chip for glucose sensing. (**A**) Overview of the different poly(dimethylsiloxane) (PDMS) layers for the tissue chamber unit. Scale bar = 5 mm. (**B**) Microfluidic system for the glucose-sensing chamber. The electrode was placed in the center of the system in the punched hole. (**C**) Overview of the muscle chip system showing the various layers of the tissue chamber as well as the connection to the glucose-sensing chamber. (**D**) Schematic drawing and picture of the customized electrode based on glucose oxidase fabricated to sense glucose. RE = reference electrode, CE = counter electrode, WE = working electrode.

poly(dimethylsiloxane) (PDMS) (*Figure 6A*). The glucose-sensing chamber was a single PDMS layer bonded to a glass slide by air plasma (*Figure 6B*). Fluid connection was achieved by punching holes using biopsy punchers and inserting tubes in the portholes (*Figure 6C*). A customized electrode was fabricated by threading a working electrode (platinum wire, Ø 51 µm, Teflon coated) and a reference electrode (silver wire, Ø 75 µm, Teflon coated) in the lumen of an 18 G syringe needle and embedding them in fluid epoxy. A counter electrode was attached to the metal of the needle by silver paste (*Figure 6D*). On the experimental day, the electrode was carefully polished using fine sand paper and aluminum oxide slurry (0.05 µm particles) and a layer of chloride was deposited on the electrode by immersing it in 3 M KCl and exposing it to six current steps consisting of −20 µA for 1 s followed by 20 µA for 9 s. The working electrode was cleaned electrochemically in 0.1 M $H_2SO_4$ by running 10 cyclic voltammogram cycles. Next, to form an exclusion membrane on the sensor, a layer of poly-(m-phenylenediamine) (m-PD) was electropolymerized on to the working electrode by applying 20 s of 0.0 V, 5 min at 0.7 V and at 0.0 V. Finally the sensor was modified by addition of glucose oxidase by embedding a PBS solution consisting of glucose oxidase (60 mg ml$^{-1}$), BSA (30 mg ml$^{-1}$), and

poly(ethylene glycol) diglycidyl ether (60 mg ml⁻¹) and 2% glycerol on top of the electrode via 2 hr incubation at 50°C.

In parallel, FDB fibers were isolated as described but cultured on 4-mm paper patches (filter 114, Whatman) by diluting the fiber solution 5:1 in extracellular matrix gel and adding 50 µl to each patch. Two hours prior to experiment, the fibers were starved from serum and glucose. Just before experiment start, the microfluidic system was assembled. First, the electrode was inserted into the detection chamber and calibrated by perifusing solutions with known glucose concentrations through the system. Next, the fiber-containing paper patch was inserted into the tissue chamber and perifused with serum-free Dulbecco´s minimal essential medium (DMEM) containing 4 mM glucose for 3–5 min. Then, the glucose concentration in the perifusate was monitored during basal and insulin-stimulated conditions and Δglucose was calculated as previously described (*Trouillon et al., 2017*). Colchicine and Noco treatment for 2 hr was achieved by pre-incubating the paper patches containing fibers in colchicine and Noco and keeping the drugs in the perifusate at all times after the assembly of the tissue chamber. Noco treatment for 5 min was achieved by switching the perifusate to one containing Noco 5 min prior to insulin stimulation.

## Western blotting

Samples were lysed in lysis buffer (50 mM Tris base, 150 mM NaCl, 1 mM Ethylenediaminetetraacetic acid (EDTA), 1 mM ethylene glycol-bis(β-aminoethyl ether)-N,N,N′,N′-tetraacetic acid (EGTA), 50 mM sodium fluoride, 5 mM pyrophosphate, 2 mM sodium orthovanadate, 1 mM dithiothreitol, 1 mM benzamidine, 0.5% protease inhibitor cocktail [Sigma P8340], 20% NP-40, pH 7.4) before processing as previously described (*Knudsen et al., 2020a*). The following antibodies were used: phospho (p)-Akt Thr308 (9275, CST), Akt (9272, CST), p-TBC1D4 Thr642 (4288, CST), or TBC1D4 (ab189890, Abcam). Coomassie staining was used as a loading control.

## Immunolabeling

Human fiber bundles were teased into individual fibers and transferred to wells in a 24-well plate containing PBS using fine forceps. FDB fibers were similarly incubated in PBS. Fibers were washed 3 × 10 min in PBS and incubated in blocking buffer 1% bovine serum albumin (Merck), 5% goat serum (16210-064, Gibco), 0.1% Na Azide (247-852, Merck), 0.04% Saponin (27534-187, VWR) for 1 hr. The muscle fibers were then incubated in blocking buffer containing primary antibodies overnight at 4°. The next day, the fibers were washed 3 × 10 min in PBS containing 0.04% saponin and incubated in blocking buffer with Alexa 488 anti-rabbit or Alexa 568 anti-rabbit or anti mouse (Invitrogen) for 2 hr. Finally, the fibers were washed 3 × 10 min in PBS and mounted on glass slides in Vectashield (H-1000, Vector Laboratories) or imaged directly from the glass bottom dish. The following antibodies were used, raised in rabbit: GLUT4 (PA5-23052, Invitrogen), detyrosinated α-tubulin (AB48389, Abcam), Syntaxin6 (110 062, Synaptic Systems), or in mouse: GLUT4 (MAB8654, R&D Systems), α-tubulin (T9026, Merck).

## Image acquisition and processing

Imaging was performed using the following systems: Zeiss 710, 780, 900, 980 or Elyra PS.1. Confocal imaging was performed using a Zeiss Plan-Apochromat ×63 1.4 NA objective. Laser source was an Argon laser emitting excitation light at 488 nm (25 mW) and helium neon laser emitting excitation light at 543 nm (1.2 mW), assembled by Zeiss. Emission light was collected by PMTs with matching beam splitters by Zeiss. The different channels were acquired sequentially. All live-imaging was performed in an integrated incubator at 37° in 5% $CO_2$ and the fibers were kept in α-MEM containing drug/hormone as described. Specific imaging details for time series are provided in figure legends. Time series with 1 image per second for 60 s were obtained for mCherry-Tubulin and GLUT4-GFP dual color imaging. EB3-GFP time series were 1 image per second for 30 or 60 s. GLUT4-GFP imaging were time series of 30–300 s with 0.1–1 image per second. At all times, pixel size was kept at ≤90 × 90 nm. The pixel dwell time was 1.27 µs. To visualize dynamics, color-coded projections were generated as described in figure legends. In these projections, moving objects appear rainbow colored whereas static objects appear white.

Structured Illumination Microscopy (SIM) was performed using an Elyra PS.1 system (Zeiss), with a Zeiss Plan-Apochromat ×63 1.4 NA objective and a ×1.6 tube lens. The system was driven by Zen

Black 2.3 SP1 from Zeiss which automatically assigns a diffraction pattern for each used wavelength (namely 28 μm for 488 nm, and 34 μm for 561 nm). Laser source was diode lasers, emitting excitation light at 488 nm (200 mW) and 561 nm (200 mW), assembled by Zeiss. Emission light was collected by a PCO.edge 5.5 sCMOS camera (PCO, Kelheim, Germany) with matching beam splitters by Zeiss. The different channels were acquired sequentially. Settings for image collection aimed at obtaining a dynamic range >5000 grayscale levels, and bleaching was assessed to be <20% of the dynamic range across the imaging sequences. 3D stacks were acquired at 100 nm steps by using a PI E-655 Z-piezo drive by Physik Instrumente (Karlsruhe, Germany).

Unless otherwise noted, images shown are single frames. For visualization purposes only, some images were cropped and contrast or levels were linearly adjusted. Images were processed using ImageJ (*Tong et al., 2001*) and Adobe Photoshop 21.2.1 (Adobe).

### 3D-SIM image reconstruction

The structured illumination image processing module built in Zen Black 2.3 was used for reconstruction, keeping a theoretical PSF and the same noise filter parameter (−5.5) for all the processed images. The resulting super-resolution images were kept in raw scaling and were baseline shifted. Super-resolution image quality was assessed by applying FFT to the reconstructed images, compared to FFT of the widefield acquisition. System performance was assessed by using an Argolight SIM patterned standard sample (Argolight, Pessac, France), obtaining resolutions ~120 nm consistently, and PSF was evaluated using 100 nm TetraSpeck beads from Zeiss.

### Live-imaging videos

Representative videos were generated from the live-imaging time series at 10 frames per second. For GLUT4-GFP and mCherry-Tubulin dual color time series (*Figure 1—video 1*), the video was generated by merging the two channels with the GLUT4-GFP channel being green and the tubulin channel being magenta. To facilitate visualization, the single-color time series (*Figure 1—video 2*, *Figure 2—video 1*, *Figure 5—video 1*) were generated by removing every other frame and switching the colors from green to red between remaining frames.

### Depletion and relocalization analysis

Using the particle analysis tool in ImageJ (*Schneider et al., 2012*), GLUT4-GFP structures (sized >0.02 $\mu m^2$, circularity between 0 and 1) from background and threshold adjusted 8-bit images were identified and structure areas were determined. From this, small structures <0.04 $\mu m^2$ were counted and related to the total number of structures as a reference for overexpression. These structures were sized from the smallest resolvable and up to ~225 nm in diameter. We analyzed this fraction since insulin induces membrane insertion of small 50–70 nm GSVs (*Martin et al., 1997*) which are expected to be part of this fraction. For relocalization analysis, structures were identified as for the depletion analysis. The individual structures were allocated into one of the following three groups: large structures (>4 $\mu m^2$), medium-sized structures (between 0.4 and 4 $\mu m^2$), and small structures (<0.4 $\mu m^2$).

### Polymerization rate analysis

Via calponin homology domains, EB3 proteins interact with tubulin at the microtubule tip and can thus be used to label polymerizing microtubules (*Stepanova et al., 2003*). On 8-bit threshold-adjusted images, the tip of polymerizing microtubules were identified as a 0.08–0.2 $\mu m^2$ region with a circularity between 0.2 and 1 with accumulated EB3-GFP signal. Based on these criteria, the average number of polymerizing microtubules was calculated per image in a 30- or 60-s time series with an image every second.

*Tracking analysis* was performed on 8-bit images using the ImageJ plug in TrackMate. Threshold was adjusted in TrackMate and tracking settings were adjusted to maximize fitting of the automated tracking. The following tracking settings were used, for GLUT4-GFP: LoG detector, 0.5 μm blob diameter estimate, LAP tracker, 1.5 μm for gap-closing and max. distance, and a max. frame gap of 1. Settings were similar for EB3-GFP, except 2 μm was used as gap-closing distance, max. distance was 1 μm and max. frame gap was 2. Tracks with a <1.5 μm displacement was not considered microtubule-based GLUT4 movement or microtubule polymerization and excluded from further analysis.

## Statistical analyses

Results are shown as mean, mean with individual values or mean ± SD. Statistical testing was performed using t-test or one- or two-way analysis of variance (ANOVA as described in the figure legends. Tukey's post hoc test was performed following ANOVA). The significance level was set at $p < 0.05$.

## Acknowledgements

We acknowledge the Core Facility for Integrated Microscopy, Faculty of Health and Medical Sciences, University of Copenhagen and especially Pablo Hernández-Veras for his guidance with the SIM imaging. We acknowledge Prof. Johan Auwerx (École Polytechnique Fédérale de Lausanne) for help with the mouse work. Finally, we acknowledge Profs. Bente Kiens and Erik A Richter (Department of Nutrition, Exercise and Sports, University of Copenhagen) for the expertise and help with conducting the human study. This study was financed by grants to TEJ (Novo Nordisk Foundation [NNF] Excellence project #15182), Lundbeck Foundation (LF) Ascending Investigator (R313-2019-643), to JFPW (NNF16OC0023046, LF R266-2017-4358 and the Danish Research Medical Council FSS8020-00288B), to JRK (a research grant from the Danish Diabetes Academy [DDA], which is funded by the NNF, NNF17SA0031406 and an International Postdoc grant from the Independent Research Fund Denmark, #9058-00047B), to CHO (Postdoc research grant from DDA, #NNF17SA0031406).

## Additional information

### Competing interests

Jonas R Knudsen, Janne R Hingst: Affiliated with Novo Nordisk A/S. Jørgen FP Wojtaszewski: has ongoing collaborations with Pfizer inc and Novo Nordisk A/S unrelated to this study. The other authors declare that no competing interests exist.

### Funding

| Funder | Grant reference number | Author |
|---|---|---|
| Novo Nordisk Fonden | 15182 | Thomas Elbenhardt Jensen |
| Novo Nordisk Fonden | 16OC0023046 | Jørgen FP Wojtaszewski |
| Novo Nordisk Fonden | 17SA0031406 | Jonas R Knudsen |
| Lundbeckfonden | R313-2019-643 | Thomas Elbenhardt Jensen |
| Lundbeckfonden | R266-2017-4358 | Jørgen FP Wojtaszewski |
| Sundhed og Sygdom, Det Frie Forskningsråd | FSS8020-00288B | Jørgen FP Wojtaszewski |
| Sundhed og Sygdom, Det Frie Forskningsråd | #9058-00047B | Jonas R Knudsen |
| Danish Diabetes Academy | NNF17SA0031406 | Carlos Henriquez-Olguin |
| Independent Research Fund Denmark | #9058-00047B | Carlos Henriquez-Olguin |

The funders had no role in study design, data collection, and interpretation, or the decision to submit the work for publication.

### Author contributions

Jonas R Knudsen, Conceptualization, Resources, Formal analysis, Supervision, Funding acquisition, Validation, Investigation, Visualization, Methodology, Writing – original draft, Project administration, Writing – review and editing; Kaspar W Persson, Carlos Henriquez-Olguin, Validation, Investigation, Writing – review and editing; Zhencheng Li, Nicolas Di Leo, Sofie A Hesselager, Steffen H Raun, Janne R Hingst, Martin Wohlwend, Jørgen FP Wojtaszewski, Investigation, Writing – review and editing; Raphaël Trouillon, Martin AM Gijs, Supervision, Writing – review and editing; Thomas Elbenhardt

Jensen, Conceptualization, Resources, Supervision, Funding acquisition, Investigation, Writing – original draft, Project administration, Writing – review and editing

### Author ORCIDs
Jonas R Knudsen ![ORCID] http://orcid.org/0000-0002-5471-491X
Nicolas Di Leo ![ORCID] http://orcid.org/0000-0002-6268-890X
Thomas Elbenhardt Jensen ![ORCID] http://orcid.org/0000-0001-6139-8268

### Ethics
The work involving human subjects was approved by the Copenhagen Ethics Committee (H-6-2014-038; Copenhagen, Denmark) and complied with the guidelines of the 2013 Declaration of Helsinki. Informed written consent was obtained from all subjects prior to entering the study.

All animal experiments were approved by the Danish Animal Experimental Inspectorate or by the local animal experimentation committee of the Canton de Vaud under license 2890 and complied with the European Union legislation as outlined by the European Directive 2010/63/EU. The current work adheres to the standards outlined in the ARRIVE reporting guidelines.

### Decision letter and Author response
Decision letter https://doi.org/10.7554/eLife.83338.sa1
Author response https://doi.org/10.7554/eLife.83338.sa2

## Additional files

### Supplementary files
• MDAR checklist

### Data availability
All data generated or analyzed during this study are included in the manuscript and supporting files.

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
