## [Editor Report]

This manuscript reveals localization of Glut4 glucose transporters at microtubules in mouse and human muscle fibers and shows that disruption of microtubules or a kinesin^-1^ motor alters Glut4 trafficking. Evidence is also provided supporting the idea that insulin resistance disrupts Glut4 dynamics at microtubules. Overall, these studies provide compelling evidence that Glut4 and its regulation by insulin involves Glut4 movements that require microtubule function.

---

## [Decision Letter]

**Decision letter after peer review:**

Thank you for submitting your article "Microtubule-mediated GLUT4 trafficking is disrupted in insulin resistant skeletal muscle" for consideration by *eLife*. Your article has been reviewed by 2 peer reviewers, and the evaluation has been overseen by a Reviewing Editor and Anna Akhmanova as the Senior Editor. The following individuals involved in review of your submission have agreed to reveal their identity: Amira Klip (Reviewer #3).

Essential revisions:

1) This issue of microtubules being required for proper localization of Glut4 into an insulin sensitive compartment is viewed as important, and the data are supportive. However, there are further experiments that should be performed to test or strengthen this point (see Reviewer 2 comments on first weakness, and Reviewer 3 comments on Figure 2 in particular).

2) The Kinesore data do not appear to support the authors' hypothesis (see Reviewer 2 comments in particular). Please address this issue in a substantive way, with additional data or commentary and appropriate interpretations.

*Reviewer #1 (Recommendations for the authors):*

Perhaps a better way to probe for the effects of insulin on GLUT4 dynamics would be to use the FRAP assay they employed to measure anterograde and retrograde trafficking in basal condition (FigS1G-H). This assay may be a more robust means of capturing the effects of insulin on GLUT4 dynamics; doing so would greatly strengthen the manuscript.

The manuscript would benefit from more detailed discussion of their results and perhaps editing of the amount of data presented. Perhaps too much narrative has been removed in trying to aware of the overall the length of the text. There are too many instances where data are presented as an observation but not fully explored. As an example, the statement on line 94-95 is too vague/descriptive as are the data in Figure 1SD. In my view, focusing the manuscript by removing extraneous data and more clearly describing the findings and how they advance our understanding of GLUT4 in muscle will enhance the impact of the work.

*Reviewer #2 (Recommendations for the authors):*

Overall, there are a number of technical questions (including better analysis of the colocalization of GLUT4 with Syntaxin 6) and throughout there is marked need to include the control of Noco treatment in absence of C2 or insulin.

Figure 1:

– A and B, nice parallel between mouse and human morphology and microtubule structural similarities. Please comment on fiber types – it appears that the general structure is not dependent on fiber type, but future phenotypes may be explained by fiber type, so it would be useful to introduce this variable early on, or at least in the discussion. Also, please clarify that FDB is used because its amenability for fiber isolation, and discuss how it compares to the intact muscles used.

Figure 1 – —figure supplement 1 – G and H = Y axis states "normalized" – please specify normalized to what, total fluorescence intensity?

Figure 2:

– Data describing GLUT4 dependence on microtubules for traffic is sound, but the dependence of insulin-stimulated GLUT4 traffic on intact microtubules is lacking the essential control of the effect of Noco in the basal state (A-C).

– While in aggregate the results support the concept that GLUT4 traffic along microtubules is required for proper GLUT4 sortin, but not for acute insulin mobilization of GLUT4 from GSV. However, testing the effect of insulin in all conditions would go a long way to clarify this concept, if possible.

– Is the effect of Noco on the insulin response (minute) seen also with other stimuli? AICAR? Contraction? It may be that this is work for a future study, and could be commented so.

– Please consider to stimulate with insulin to mobilize GLUT4, then disrupt microtubules and perform a re-stimulation with insulin – in this way both acute (1 stimulation) and prolonged (re-stimulation) insulin-regulated GLUT4 traffic along microtubules could be examined

– B-C, Are the data in C the results from 4 experiments, as performed in B? It seems somewhat misleading if the variability within a single experiment is as shown in B, but when pooled in C the effect is now apparent when there was no significant insulin effect in any single experiment. If C is indeed an average of 4 independent experiments, why is only a single experiment shown in B?

– Please clarify the sources of data for B and C, and include error bars for both B and C. Also specify in C, normalized to what?

– D-E, 15 hours Noco, then 9 hours recovery – why these hours? If previous experiment used 4 hours, why are 15 hours required to disrupt microtubules in this experiment?

o Is there a reason why this experiment was not also conducted in the presence of insulin? Similarly, in F-G, would this localization change in presence of insulin?

– Please indicate "mouse" in G

– In supplement 1 to this figure:

– B-C, Normalized to what?

– D , Why was basal+Noco not tested here but in E-G insulin is not tested?

– D, Noco treatment back to 4 hours but in F-H treatment is 15 hours please state the rationale for treatment times

– It appears the Pearson's R value for Syntaxin 6 and GLUT4 was applied to the entire fiber, or is it the Entire image? Fields? In any case, it would be highly useful to segregate colocalization according to depots of GLUT4: at nucleation site clusters, intermediate structures and small structures. This could be far more informative than the overall which will likely muddle any potential colocalization. Additionally, since these are overexpressed proteins, it is important to calculate the colocalization of one protein within the other's pool and vice versa.

Figure 3:

– A-B, glucose uptake phenotype also nicely demonstrated for FDB in C-F. Please suggest explanation for why this insulin effect was not apparent in Figure 2?

– In figure 2, could endogenous GLUT4 be detected in static images to examine localization relative to microtubules in soleus or EDL muscles? This could provide some context as to fiber type specific dependence on microtubules for insulin-mediated GLUT4 traffic.

– Noco treatment only 2 hours, compared to 4-15 hours when measuring GLUT4 traffic , what is the rationale for this difference?

– Please provide potential explanations for why for microtubules are important for the acute insulin action on on 2-DG uptake but not GLUT4 traffic. This is a key question that must be addressed.

– H, What is different from Figure 2E – in 3H the number of small structures is elevated with 2 hours Noco but in 1E they are reduced. Does the presence of insulin alter this distribution?

– G-H, What does GLUT4 localization look like in presence of insulin +/- Noco?

Figure 4

– Nice demonstration of Kinesin-1 dependence. In C, It would be useful to see this performed with insulin +/- Noco and present in Figure 2

Figure 5:

– 5B and supplement 1B, why is there no basal + C2? If the conclusion is that C2 inhibits insulin stimulated vesicle movement, it is essential to show that this is not happening in the basal state

Discussion:

– The finding that more GLUT4 'vesicles' appear in intermediate and small structures along the microtubule in response to Noco is fascinating. Given that those pools increase, they are unlikely to be the ones responding to insulin, unless they no longer engage with the elements that cause mobilization (whether Kinesin-1 or other). On the other hand, Kinesore causes GLUT4 dispersion into small clusters and promotes translocation to the membrane even additive to insulin's action. Please reconcile these observations.

– The dispersion of GLUT4 upon activation of Kinesin-1 via Kinesore is also very interesting, but requires further assessment of specificity towards Kinesin-1.

– One cannot call the 'puncta' vesicles, these are likely aggregates unless the point spread function is calculated for each item and found to be individual

– Throughout the study there is no mention of the other fiber and motors that contribute to GLUT4 traffic in muscle. In the past the authors have not detected insulin-dependent remodeling of transfected actin, but this does not rule out a dependency of the positioning of the GLUT4 pools on actin cytoskeletal elements. -

– Key and related to the above is testing whether Noco for 2 h alters any of the known elements of the Rac1 signaling pathway or the proteins regulating actin filaments. There are studies in the literature showing interconnectivity between the microtubule and microfilament networks.

---

## [Author Response]

Essential revisions:1) This issue of microtubules being required for proper localization of Glut4 into an insulin sensitive compartment is viewed as important, and the data are supportive. However, there are further experiments that should be performed to test or strengthen this point (see Reviewer 2 comments on first weakness, and Reviewer 3 comments on Figure 2 in particular).2) The Kinesore data do not appear to support the authors' hypothesis (see Reviewer 2 comments in particular). Please address this issue in a substantive way, with additional data or commentary and appropriate interpretations.

Thank you for the comprehensive and constructive feedback on our work. Based on the comments from the reviewers and the reviewing editor, and with particular focus on the above points, we have performed additional experiments to better understand the connection between insulin-regulated GLUT4 localization and to characterize the effect of kinesore in adult muscle in more detail. Our new findings are integrated into an updated and improved version of the manuscript. We address the reviewer comments point by point below.

Reviewer #1 (Recommendations for the authors):Perhaps a better way to probe for the effects of insulin on GLUT4 dynamics would be to use the FRAP assay they employed to measure anterograde and retrograde trafficking in basal condition (FigS1G-H). This assay may be a more robust means of capturing the effects of insulin on GLUT4 dynamics; doing so would greatly strengthen the manuscript.

Thank you for this proposal. We agree that FRAP experiment would be a good idea and may be a more robust means of capturing an insulin effect. Unfortunately, the first author of this paper is no longer employed in the lab and thus it is currently not feasible to perform this experiment.

Worth mentioning, we previously attempted to isolate an insulin effect on anterograde kinesin-dependent trafficking by using the cytoplasmic dynein inhibitor Ciliobrevin D. However, we could also not detect any effect of insulin using this approach.

The manuscript would benefit from more detailed discussion of their results and perhaps editing of the amount of data presented. Perhaps too much narrative has been removed in trying to aware of the overall the length of the text. There are too many instances where data are presented as an observation but not fully explored. As an example, the statement on line 94-95 is too vague/descriptive as are the data in Figure 1SD. In my view, focusing the manuscript by removing extraneous data and more clearly describing the findings and how they advance our understanding of GLUT4 in muscle will enhance the impact of the work.

We have modified the manuscript to more clearly focus on the data supporting our main conclusions. However, we also believe that publishing descriptive data and odd observations rather than just a polished storyline increases the opportunities for others to interpret and build on the work. Thus, we still include descriptive/observational data in this updated version of the manuscript.

Reviewer #2 (Recommendations for the authors):Overall, there are a number of technical questions (including better analysis of the colocalization of GLUT4 with Syntaxin 6) and throughout there is marked need to include the control of Noco treatment in absence of C2 or insulin.Figure 1:– A and B, nice parallel between mouse and human morphology and microtubule structural similarities. Please comment on fiber types – it appears that the general structure is not dependent on fiber type, but future phenotypes may be explained by fiber type, so it would be useful to introduce this variable early on, or at least in the discussion. Also, please clarify that FDB is used because its amenability for fiber isolation, and discuss how it compares to the intact muscles used.Figure 1 – —figure supplement 1 – G and H = Y axis states "normalized" – please specify normalized to what, total fluorescence intensity?

Thanks. The rationale behind choice of muscle and their fiber type distribution has been addressed (line 78-81). Normalization was min-max normalization, which is now specified in the legend.

Figure 2:– Data describing GLUT4 dependence on microtubules for traffic is sound, but the dependence of insulin-stimulated GLUT4 traffic on intact microtubules is lacking the essential control of the effect of Noco in the basal state (A-C).

The microtubule disruption by nocodazole in the insulin-stimulated state completely blocked all microtubule-mediated GLUT4 movement. This is clear from figure 2 A&B and particularly from the uploaded video showing no microtubule-mediated GLUT4 trafficking following 2h of microtubule disruption. Thus, a further reduction in GLUT4 movement without insulin stimulation is not possible. For this reason, we believe the proposed Noco treated basal fiber control to be redundant.

– While in aggregate the results support the concept that GLUT4 traffic along microtubules is required for proper GLUT4 sortin, but not for acute insulin mobilization of GLUT4 from GSV. However, testing the effect of insulin in all conditions would go a long way to clarify this concept, if possible.

For practical reasons, we are limited in the experiments we can currently perform in the lab. Thus, additional GLUT4-GFP electroporation and fiber isolation experiments are not possible to perform within a reasonable time-frame.

As a proxy measure of GLUT4 mobilization from GSV, we tested whether insulin would reduce the number of small vesicles (<0.4 µm2) in our live imaging experiments shown in figure 2A-B. However, we could not detect any significant insulin regulation by this approach.

– Is the effect of Noco on the insulin response (minute) seen also with other stimuli? AICAR? Contraction? It may be that this is work for a future study, and could be commented so.

This is of course an interesting and important question. We have not studied this. In the new experiments performed in our revision process, we observed a marked increase in AMPK activation (p-AMPK and p-ACC) following kinesore treatment, both in basal and insulin stimulated EDL and soleus muscle. Despite this AMPK activation, we observed no increase in glucose uptake in the basal state and impaired insulin-stimulated glucose uptake. The prediction from these data would be that the effect of Noco/kinesore would also be seen with AICAR/AMPK activation and contractions.

– Please consider to stimulate with insulin to mobilize GLUT4, then disrupt microtubules and perform a re-stimulation with insulin – in this way both acute (1 stimulation) and prolonged (re-stimulation) insulin-regulated GLUT4 traffic along microtubules could be examined

We stimulated L6 cells with insulin (100nm, 15 min) before 2h ± Noco or Colchicine and then performed re-stimulation with insulin or kept cells basal. There was a clear main effect of both Noco and Colchicine when cells were pre-treated with insulin. Overall, there was no clear effect of microtubule disruption on insulin-stimulated GLUT4 translocation when cells were pre-stimulated with insulin before microtubule disruption. From these data, it seems that, in the L6 cells overexpressing GLUT4, a residual insulin-responsive GLUT4 pool remains after microtubule disruption, even with prior GLUT4 mobilization by insulin.

– B-C, Are the data in C the results from 4 experiments, as performed in B? It seems somewhat misleading if the variability within a single experiment is as shown in B, but when pooled in C the effect is now apparent when there was no significant insulin effect in any single experiment. If C is indeed an average of 4 independent experiments, why is only a single experiment shown in B?

Yes, data are from 4 experiments performed as in B. Each data point set reflects the average effect size from each experiment. In addition to the experiment shown in B, the data in C also originate from the basal and insulin control groups in figure 5.

Because this particular figure panel was highlighted as confusing by two reviewers and since we could not detect an effect of insulin in single experiments, we decided to remove the panel from the manuscript. We discuss several possibilities for the lack of significant insulin effect on GLUT4 movement in individual experiments in the Discussion section (lines 342 to 361).

– Please clarify the sources of data for B and C, and include error bars for both B and C. Also specify in C, normalized to what?

See reply above.

– D-E, 15 hours Noco, then 9 hours recovery – why these hours? If previous experiment used 4 hours, why are 15 hours required to disrupt microtubules in this experiment?

For practical reasons, we choose to do an overnight treatment with nocodazole, then image the fibers in the morning, allow recovery throughout the working day and re-image the fibers again in the afternoon. This amounted to 15h Noco treatment and 9h recovery. As seen from the representative live-imaging of the experiment in 2A-B, the changes induced by microtubule disruption had occurred already after 4h. Hence, we do not believe that a prolonged 15h disruption would be required for our localization observations. However, as also noted by the reviewer in a later comment, the number of small GLUT4 membrane structures was higher after 2h of nocodazole treatment compared to the control condition while it was lower after 15h. We have not investigated this time-dependent effect further.

– Is there a reason why this experiment was not also conducted in the presence of insulin? Similarly, in F-G, would this localization change in presence of insulin?

The research question we asked in this experiment was whether microtubule disruption would eventually result in GLUT4 mis/re-localization and whether this was reversible.

Whether insulin stimulation would affect mis/re-localized GLUT4 is an interesting question, but would in our opinion have been better addressed by an isolated experiment.

For F-G, our hypothesis was that insulin would reduce GLUT4 Syntaxin6 co-localization. Since the correlation between GLUT4 and Syntaxin6 was low already in the basal state and unaffected by nocodazole we did not pursue this hypothesis further.

– Please indicate "mouse" in G

Corrected.

– In supplement 1 to this figure:– B-C, Normalized to what?

We normalized the individual intensity to the average intensity across the full data set.

– D , Why was basal+Noco not tested here but in E-G insulin is not tested?

The reasoning for the different treatments is similar in figure 2 and in the supplement to figure 2.

– D, Noco treatment back to 4 hours but in F-H treatment is 15 hours please state the rationale for treatment times

The treatment times were different for practical reasons outlined above.

– It appears the Pearson's R value for Syntaxin 6 and GLUT4 was applied to the entire fiber, or is it the Entire image? Fields? In any case, it would be highly useful to segregate colocalization according to depots of GLUT4: at nucleation site clusters, intermediate structures and small structures. This could be far more informative than the overall which will likely muddle any potential colocalization. Additionally, since these are overexpressed proteins, it is important to calculate the colocalization of one protein within the other's pool and vice versa.

Thank you for pointing this out. We have reanalysed the data specifically analysing the perinuclear region and using Manders´ correlation to estimate the fraction of GLUT4 in Syntaxin 6 positive regions. We did not see any changes in overlap by Noco treatment (0.269 vs. 0.273, p=0.96).

Figure 3:– A-B, glucose uptake phenotype also nicely demonstrated for FDB in C-F. Please suggest explanation for why this insulin effect was not apparent in Figure 2?

Please see our response to weakness 1 by reviewer #2.

– In figure 2, could endogenous GLUT4 be detected in static images to examine localization relative to microtubules in soleus or EDL muscles? This could provide some context as to fiber type specific dependence on microtubules for insulin-mediated GLUT4 traffic.

Indeed, this would be possible to do. However, our ex vivo incubation glucose uptake data suggest no major differences between EDL and Soleus. For this reason, we would expect no major fiber type differences and did not prioritize these analyses.

– Noco treatment only 2 hours, compared to 4-15 hours when measuring GLUT4 traffic , what is the rationale for this difference?

Our ex vivo preparations are difficult to keep viable for extended time periods. Since we hypothesized that the immediate insulin-induced glucose uptake would be preserved whereas longer duration microtubule disruption would be inhibitory, we decided on a long ex vivo incubation time at which the muscles would still be viable, i.e. 2h.

We judged that the additional information from re-performing the isolated FDB fiber experiments conducted in figure 1 and 2 using only 2h Noco would be limited and thus decided not to perform these experiments.

– Please provide potential explanations for why for microtubules are important for the acute insulin action on on 2-DG uptake but not GLUT4 traffic. This is a key question that must be addressed.

We are unsure whether there has been a mistake in posing this comment as we see the opposite, i.e. that microtubules are important for GLUT4 traffic but not acute glucose uptake. Our working hypothesis is that the muscle fiber retains a GLUT4 reservoir that is capable of reaching the surface membrane even in the absence of microtubules, likely from a pre-tethered pool as proposed by Samuel Cushman’s group (PMID: 22297303). This reservoir may allow the muscle fiber to retain a normal initial insulin response on glucose uptake. We discuss this hypothesis in the discussion (lines 308-321).

If this question instead relates to the fact that insulin-stimulated glucose uptake is impaired after 2h of microtubule disruption despite no effect of insulin on microtubule-mediated GLUT4 trafficking, then our response is the following:

In addition to the 2-DG uptake measurements ±Noco, we have now demonstrated that Kinesore treatment (2h, 50uM) also impairs insulin-stimulated 2-DG uptake in adult skeletal muscle, overall supporting the requirement of microtubule-mediated transport for 2-DG uptake. Additionally we have clearly demonstrated that the microtubules are important for adult muscle GLUT4 traffic. As pointed out, we have not demonstrated that the microtubules are important for the acute action of insulin on GLUT4 traffic in muscle fibers. We hypothesize that a fraction of the insulin-regulated GLUT4 trafficking is dependent on intact microtubules, but we currently do not have sensitive enough tools to fully test this hypothesis as discussed in the current version of the manuscript (lines 341 to 361).

– H, What is different from Figure 2E – in 3H the number of small structures is elevated with 2 hours Noco but in 1E they are reduced. Does the presence of insulin alter this distribution?

We have not investigated whether insulin would alter this distribution, nor have we investigated why there is a difference between the two time points. It would be really interesting to explore these observations further, especially since the small structures would presumably include insulin-responsive vesicles. However, as also pointed out by reviewer #2, the data would be hard to interpret without identification by compartment specific markers. Since this would require a lot of optimization to set up, we find this to lie beyond the scope of the current study.

– G-H, What does GLUT4 localization look like in presence of insulin +/- Noco?

We quantified the number of small GLUT4 structures following insulin stimulation +/- Noco in the live-imaging experiment in Figure 2, but we did not observe significant changes (see figure 4 in this rebuttal). These data were not included in this manuscript.

We believe that the lack of effect is due to lack of sensitivity in the image analyses. Supporting this notion, we previously demonstrated, that insulin-stimulated GLUT4 translocation can occur without discernible visual GLUT4 localization changes in mouse muscle (PMID: 30710396), indicating that higher resolution microscopy and more specific image analyses approaches may be a requirement.

Figure 4– Nice demonstration of Kinesin-1 dependence. In C, It would be useful to see this performed with insulin +/- Noco and present in Figure 2

During the revision work, we observed that kinesore impairs insulin-stimulated glucose uptake in incubated mouse muscle and primary human myotubes. This has made us unsure of the interpretation of the data presented in figure 4C and, also considering also reviewer 2´s comments about too many underexplored observations, we thus chose to remove it from this version of the manuscript.

Figure 5:– 5B and supplement 1B, why is there no basal + C2? If the conclusion is that C2 inhibits insulin stimulated vesicle movement, it is essential to show that this is not happening in the basal state

We retrospectively agree that this is an essential control to interpret the insulin-effect. We therefore acknowledged this limitation by changing the wording in the manuscript to reflect that we cannot conclude whether C2 inhibits the effect of insulin (lines 218-222).

Discussion:– The finding that more GLUT4 'vesicles' appear in intermediate and small structures along the microtubule in response to Noco is fascinating. Given that those pools increase, they are unlikely to be the ones responding to insulin, unless they no longer engage with the elements that cause mobilization (whether Kinesin-1 or other). On the other hand, Kinesore causes GLUT4 dispersion into small clusters and promotes translocation to the membrane even additive to insulin's action. Please reconcile these observations.

We included some speculative discussion on the identity of the structures (lines 311-318). Future studies should aim to identify the nature of these structures.

– The dispersion of GLUT4 upon activation of Kinesin-1 via Kinesore is also very interesting, but requires further assessment of specificity towards Kinesin-1.

We agree that the dispersion of GLUT4 upon kinesore treatment is very interesting but requires further assessment. We are currently uncertain of the interpretation of the dispersion data and considering also reviewer 2´s comments about too many underexplored observations, we decided to remove the data from this manuscript.

– One cannot call the 'puncta' vesicles, these are likely aggregates unless the point spread function is calculated for each item and found to be individual

We have rephrased to “structures” throughout the manuscript.

– Throughout the study there is no mention of the other fiber and motors that contribute to GLUT4 traffic in muscle. In the past the authors have not detected insulin-dependent remodeling of transfected actin, but this does not rule out a dependency of the positioning of the GLUT4 pools on actin cytoskeletal elements. -– Key and related to the above is testing whether Noco for 2 h alters any of the known elements of the Rac1 signaling pathway or the proteins regulating actin filaments. There are studies in the literature showing interconnectivity between the microtubule and microfilament networks.

Indeed our data fit well in a working model in which microtubules mediate long range movement of GLUT4 towards the cell periphery where the microtubules could be interconnected with the actin cytoskeleton that could mediate the peripheral trafficking and docking steps in the translocation process. We included a small discussion of the evidence for the involvement of actin in L6 and adult muscle and the possible crosstalk between the two cytoskeleton types in l. 400-410.